# Split RNA switch orchestrates pre- and post-translational control to enable cell type-specific gene expression

Itsuki Abe[1,2,3,4], Hirohisa Ohno[1] ✉, Megumi Mochizuki[1], Karin Hayashi[1] & Hirohide Saito ◉ [1,3] ✉

RNA switch is a synthetic RNA-based technology that controls gene expression in response to cellular RNAs and proteins, thus enabling cell type-specific gene regulation and holding promise for gene therapy, regenerative medicine, and cell therapy. However, individual RNA switches often lack the specificity required for practical applications due to low ON/OFF ratios and difficulty in finding distinct and single biomolecule targets. To address these issues, we present "split RNA switches" that integrate outputs from multiple RNA switches by exploiting protein splicing. We show that split RNA switches significantly improve the ON/OFF ratio of microRNA-responsive ON switch system by canceling leaky OFF level in human cells. Using this approach, we achieve efficient cell purification using drug-resistance genes based on endogenous microRNA profiles and CRISPR-mediated genome editing with minimal off-target-cell effects. Additionally, we demonstrate RNA-based synthetic circuits using split RNA switches to enable the detection of multiple microRNAs and proteins with logical operations. Split RNA switches highlight the potential of post-translational processing as a versatile and comprehensive strategy for advancing mRNA-based therapeutic technologies.

Regulating gene expression in response to biomolecules is a powerful strategy for monitoring intracellular environments and controlling cellular programs[1,2]. There are various examples of such gene-regulatory components in nature, represented by Tet repressors and riboswitches[2–4]. Researchers have repurposed such transcriptional or translational regulators to implement artificial gene regulatory systems in recent years to sense molecules of interest by introducing exogenous DNA or RNA elements into cells. Among such gene regulatory systems, translational regulation using exogenous messenger RNAs (mRNAs) is preferable for some medical applications to transcriptional regulation, which requires the introduction of foreign DNA, thus carrying a potential risk of non-specific genome integration. RNA switch is a synthetic mRNA-based technology that translationally controls gene expression in response to intracellular RNAs, proteins,

and small molecules[5–11]. RNA switches, particularly those targeting endogenous microRNAs (miRNAs) or proteins distinct to certain cell types, enable the control of translation for output genes specific to these cell types. Regulating output genes such as fluorescent reporter, apoptotic, and cytotoxic genes by RNA switches enables cell type-specific classification[5,6], cell-fate control[5,6,12,13], and genome editing[14]. Therefore, RNA switch technology holds great potential for gene therapy, cell therapy, and regenerative medicine.

However, existing strategies using single RNA switches targeting one molecule suffer from inadequate cell type specificity, thus hindering broad and practical applications. Target cell specificity is plagued commonly by two challenges (Supplementary Fig. 1). The first challenge is the undesired leaky expression of output protein in the OFF state ("leaky translational control"), resulting in a low ON/OFF

[1]Center for iPS Cell Research and Application, Kyoto University, 53 Kawahara-cho, Shogoin, Sakyo-ku, Kyoto, Japan. [2]Graduate School of Medicine, Kyoto University, Yoshida-Konoe-cho, Sakyo-ku, Kyoto, Japan. [3]Institute for Quantitative Biosciences, The University of Tokyo, Tokyo, Japan. [4]Department of Bioengineering School of Engineering, The University of Tokyo, Tokyo, Japan. ✉e-mail: ohno.hirohisa.3u@kyoto-u.ac.jp; hirosaito@iqb.u-tokyo.ac.jp

ratio (translation level in the ON state relative to the OFF state, i.e., signal-to-noise). Even in an "ideal" situation where only the target cells in the population possess the target miRNA activity, and the non-target cells have none, leaky translation in the OFF state of the RNA switch itself prevents efficient fluorescent protein-based classification or cell fate control gene-based purification of target cells in response to a target biomolecule. The second challenge relates to the difficulty in finding a target biomolecule whose expression level in the target cell type is so different from all the other cell types that an RNA switch can clearly distinguish the cell[15] ("lack of distinctive marker"). This issue is particularly problematic when identifying or selecting target cells from a heterogeneous population of multiple cell types. To overcome the "lack of distinctive marker" challenge, multi-input systems capable of sensing several molecules have been developed[7,15,16]. However, increased system complexity could generate noise, thus leading to a counterproductive situation where the overall ON/OFF ratio of the system decreases.

To address the "leaky translational control" and "lack of distinctive marker" challenges, we develop a mRNA-based methodology called "split RNA switch," which employs protein splicing, a post-translational modification system, to integrate translational control by multiple RNA switches. The split RNA switch simultaneously solves these two problems by (1) improving the ON/OFF ratio by suppressing leaky output protein expression in the OFF state and (2) enabling the construction of multi-input systems responding to different miRNAs and proteins. Indeed, using this system, we improve cell type specificity dramatically by increasing the ON/OFF ratio of a miRNA-responsive ON switch system from a few-fold to more than 25-fold and then, to the best of our knowledge, achieve the first demonstration of antibiotics-based purification of miRNA+ target cells solely by synthetic mRNAs. We also apply this "split ON switch" strategy to the CRISPR-Cas9 system in human cells, including induced pluripotent stem cells (hiPSCs), enabling miRNA-induced, cell type-specific genome editing with minimal off-target effects. Furthermore, we expand

the mechanism of leak cancellation to a two-output system to enhance the binariness of a miRNA-responsive toggle-like system, resulting in a clearer separation of cell types in flow cytometry two-dimensional plots. Finally, by utilising multiple split RNA switches, each targeting different miRNAs or proteins, and two orthogonal split inteins, we demonstrate a two- and three-input system capable of simultaneously sensing the combinations of different miRNAs and proteins. The split RNA switch presents a promising strategy for orchestrating pre- and post-translational regulation, which simplifies the design of RNA-based synthetic intracellular circuits and offers a comprehensive solution for accurate gene regulation using RNA switch technology.

## Results

### MicroRNA-responsive split ON switch system with enhanced ON/OFF ratio

MicroRNAs (miRNAs) are small noncoding RNAs that post-transcriptionally regulate gene expression of target mRNAs. With over 2600 miRNAs encoded in the human genome[17,18] and unique expression patterns among different cell types[19–22], miRNAs are useful markers for distinguishing cell types[15,23–27]. miRNA-sensing gene regulatory systems have been developed previously[28–30], including miRNA-responsive RNA switches[5,6] (Fig. 1a), thus allowing cell type-specific gene regulation[5,6,12–14]. We previously developed two types of miRNA-responsive RNA switches: OFF switch[5] and ON switch[6]. An OFF switch contains an antisense sequence ("miRNA target site") of the target miRNA in the 5′ UTR[5]. While it is translated in the same manner as normal mRNA in cells without the target miRNA activity (miRNA- cells), transgene expression from the OFF switch is inhibited by miRNA binding to the target site in cells possessing the specific miRNA activity (miRNA+ cells), likely due to the cleavage at the miRNA target site, followed by subsequent mRNA degradation[31] (Fig. 1a, left). Conversely, an ON switch has a miRNA target site downstream of the poly(A) tail, followed by an extra sequence in the 3′ terminus[6]. In miRNA- cells, transgene expression is inhibited by the extra sequence and a miRNA

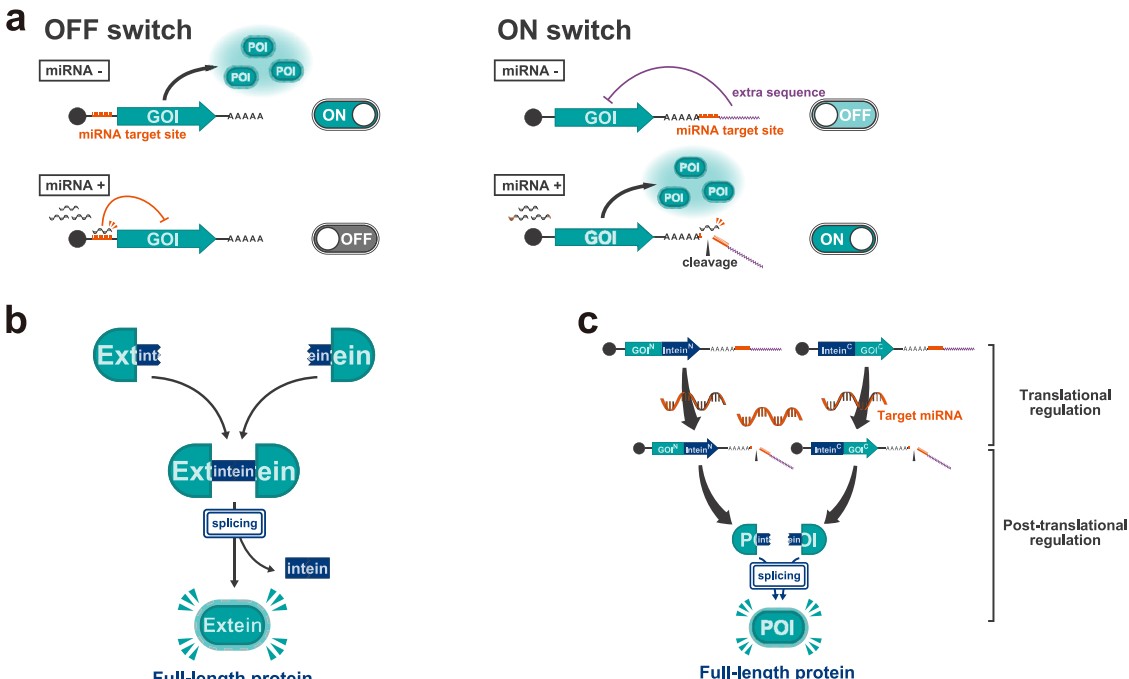

**Fig. 1 | miRNA-responsive RNA switches and *trans*-protein splicing. a** Schematic illustration of miRNA-responsive RNA OFF and ON switches coding the gene of interest (GOI). The OFF switch (left) inhibits translation of the protein of interest (POI) in the presence of miRNA activity, while the ON switch (right) promotes translation under the same conditions. **b** Schematic illustration of *trans*-protein

splicing. In this reaction, split-inteins linked to their flanking peptides (exteins) ligate post-translationally, excising themselves and seamlessly joining the exteins to generate a mature protein. **c** Schematic illustration of "split ON switch," a pair of ON switches coding protein fragments conjugated with split-inteins.

target site after the poly(A) tail. In miRNA + cells, miRNA binds to the target site, likely resulting in cleavage of the extra sequence[32] and enhancement of transgene expression from the ON switch, thus functioning as a miRNA-sensing gene activator (Fig. 1a, right). However, ON switches with different extra sequences often exhibit low ON/OFF ratios due to leaky OFF state expression, as shown in this and our prior study[6] (Supplementary Fig. 2). Therefore, substantial efforts are still required for optimal switch design to improve ON/OFF ratio for accurate control of gene expression in a target cell.

To achieve output protein control with a better ON/OFF ratio in a target cell-dependent manner without engineering the RNA switch itself, we employed protein splicing and integrated translational controls of multiple RNA switches to suppress protein activity level in the OFF state ("split RNA switch," Fig. 2a). Protein- (trans-) splicing is a type of post-translational modification that occurs between protein fragments linked to peptide sequences called split-inteins[33,34]. Corresponding split-intein segments ligate with each other and then excise themselves from the flanking sequence called extein, seamlessly joining extein sequences, and finally generating a full-length protein[35,36](Fig. 1b). The split-intein used in this study, NpuDnaE, requires only the N-terminal amino acid residue of the C-extein to be cysteine for efficient protein splicing[37,38] to allow great flexibility in extein sequences, enabling reconstitution in a wide variety of proteins with a high efficiency[37,39–43] (Supplementary Fig. 3). We designed a pair of ON switches coding protein fragments conjugated with split-inteins, termed "split ON switch" (Fig. 1c and Supplementary Fig. 4). We tested several extra sequences of miR-21-5p-responsive ON switch in the presence or absence of miR-21-5p mimic (a chemically modified double-stranded RNA that mimics endogenous miRNA) in HEK293FT cells and chose the extra sequence of the ON switch (Supplementary Fig. 2), which showed a modest ON/OFF ratio in a similar level to the previous study[6].

To improve the ON/OFF ratio of the output protein in response to target miRNA activity, we devised the split RNA switch approach based on the following mechanism (Fig. 2a). First, we introduced a pair of ON switches, coding N- and C-terminal protein fragments linked to split-inteins (split ON switch pair: GOI$^N$-Intein$^N$-ON and Intein$^C$-GOI$^C$-ON) along with an OFF switch coding an inactivated mutant C-terminal fragment linked to C-terminal split-intein (Intein$^C$-GOI$^C$m-OFF$_{21}$, hereafter referred to as "leak-canceller"). Whereas ON switches generate N- and C-terminal fragments of the protein of interest in miRNA + cells (Fig. 2a, upper right), little translation arose from the OFF switch. Then, the N- and C-terminal fragments are immediately spliced to form a full-length, functional protein. In contrast, in miRNA-cells (Fig. 2a, lower right), ON switches show leaky expression of N- and C-terminal fragments. Meanwhile, the OFF switch generates mutated C-terminal fragments. Because the mutated C-terminal fragments are present in larger amounts compared to the normal C-terminal fragments, most of the N-terminal fragments, leaked from the ON switch, are expected to undergo the irreversible splicing reaction with the inactivated C-terminal fragments. Therefore, the formation of full-length functional output proteins with a normal C-terminal fragment will be inhibited. As a result, the leaky activity of the output protein in miRNA-negative cells will be suppressed. Theoretically, the more OFF switches are introduced, the stronger this competitive cancellation effect will be.

To test our hypothesis, we performed a reporter assay in HEK293FT cells using hmAG1 as the output protein for quantification by flow cytometry and fluorescence microscopy (Fig. 2b–d). In this reporter assay, we used a miR-21-5p mimic to induce miR-21-5p activity in HEK293FT cells (miR-21-5p activity is otherwise low in HEK293FT cells[15]), and an iRFP670-coding mRNA as a reference. In cells with a negative control (NC) mimic, the single ON switch coding full-length hmAG1(AG-ON$_{21}$) exhibited over one-third of fluorescence intensity compared with cells treated with the miR-21-5p mimic,

indicating substantial leaky expression in the OFF state. Similarly, a moderate ON/OFF ratio was observed when introducing only a split ON switch pair coding N- and C-terminal fragments. Meanwhile, when transfecting the split ON switch pair in equal amounts with the leak-canceller (OFF switch coding the inactivated C-terminal fragment), the fluorescence intensity in cells without miR-21-5p mimic was reduced to about 10% of the level displayed by cells with the mimic, thus increasing the ON/OFF ratio to 10-fold. Furthermore, a split ON switch pair with four times the amount of the leak-canceller resulted in an ON/OFF ratio exceeding 25 times in response to miRNA activity. The slight decrease in ON levels in the miR-21-5p mimic condition could be attributed to leaky translation from the OFF switch. From these results, we confirmed that the split ON switch system suppresses undesired leaky output in the OFF state, thereby improving the ON/OFF ratio of the ON switch system.

## Cell-fate control using drug-resistance genes and a suicide gene

Next, to verify the versatility of our split ON switch system, we applied the system to three commonly used antibiotic resistance genes-encoding mRNAs (puromycin N-acetyltransferase (PAC) (Fig. 3a), hygromycin-B 4-O-kinase (HPH) (Fig. 3c), and blasticidin-S deaminase (BSR) (Fig. 3e)) and controlled cell fate based on miR-21-5p activity. In each experiment, we used a pair of split ON switches, coding N- and C-terminal fragments of the gene, and a leak-canceller, coding an inactivated C-terminal fragment of PAC. It is unnecessary to mutate C-terminal fragments for each gene because mutated C-terminal fragments for leakage-cancellation are required only to form an inactive protein through splicing with the N-terminal fragment. We evaluated the survival rate of HeLa cells, which have high endogenous activity of miR-21-5p when cultured in the presence of antibiotics and treated with either miR-21-5p or negative control (NC) inhibitors (Fig. 3b, d, f and Supplementary Fig. 5a–c). In all three experiments, transfection of an ON switch coding the full-length gene led to leaky survival of HeLa cells treated with miR-21-5p inhibitor and antibiotics due to translational leakage of the resistance gene. Especially in the cases of HPH and BSR ON switches, survival was nearly unaffected by miR-21-5p activity. These small changes were observed presumably because the amount of transfected mRNA was so large that the quantity of leaked resistance protein was sufficient to rescue the cells almost completely. Notably, in all three experiments, the introduction of the split ON switch pair along with the leak-canceller allowed HeLa cells with high miR-21-5p activity to survive, whereas significantly suppressing the survival of HeLa cells deprived of miR-21-5p activity under antibiotics-added culture conditions. Thus, the split RNA switch system is highly versatile in terms of its output genes, which enables cell type-specific fate control using antibiotic resistance genes optimised for target cells.

In addition to these genes for cell type-specific drug selection, we regulated herpes simplex virus 1 thymidine kinase (HSV-TK) by the split ON switch system and performed target cell (miRNA + )-specific apoptosis induction that is dependent on both target miRNAs and ganciclovir (GCV)[44] (Fig. 3g). In the HSV-TK/GCV system, ganciclovir, a thymidine analogue, is monophosphorylated exclusively by the exogenous HSV-TK[45]. Then, the monophosphate is further phosphorylated by endogenous TK to produce GCV-triphosphate. This triphosphate is incorporated into replicating DNA, terminating the elongation of DNA strands and consequently leading to apoptosis[46–48]. We treated HeLa cells with GCV at the same time as the transfection of RNA switches and miRNA inhibitors (miR-21-5p or NC inhibitors). After 24 h, we evaluated cell viability by the WST-1 assay (Fig. 3h). TK$_{A266C}$, a full-length HSV-TK with A266C, an expected single amino acid mutation introduced after protein splicing, reduced the survival rate of HeLa cells to about 15% in the presence of GCV compared to that without GCV. When using the ON switch coding TK$_{A266C}$, the survival rate of HeLa cells with miR-21-5p inhibitor declined to $30.6 \pm 6.1\%$ by

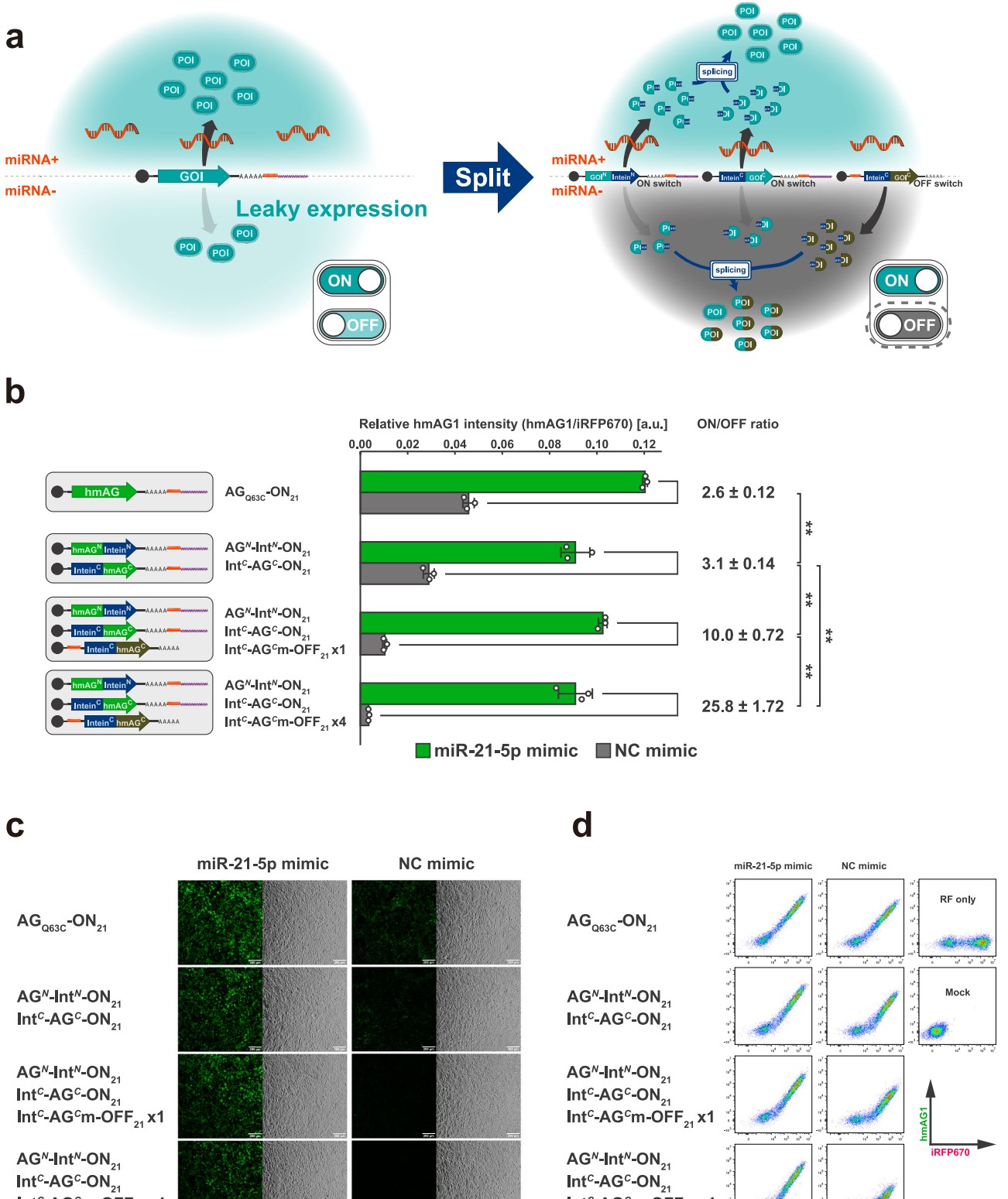

**Fig. 2 | Reporter assay of split ON switch system. a** Schematic illustration of the strategy to improve the ON/OFF ratio of ON switch systems. Introducing a "leak-canceller", an OFF switch coding an inactive C-terminal fragment, together with the split ON switch, enables the suppression of leaky protein activity in miRNA-cells and enhances the ON/OFF ratio. **b** Relative hmAG1 intensity (hmAG1/iRFP670) of HEK293FT cells treated with miR-21-5p or negative control (NC) mimics. a.u., arbitrary units. Error bars represent means ± SD ($n = 3$), and data of each biological replicate are shown as a point. Statistical analysis by two-sided Welch's $t$ test,

**$P < 0.01$. Each $P$-value is listed in Supplementary Table 4. Source data are provided as a Source Data file. **c** Representative microscopic images of HEK293FT cells transfected in the same conditions as shown in (**b**). Green fluorescence (left) and bright-field (right) images are shown for each condition. Scale bar, 200 μm. **d** Representative 2D flow cytometry plots. The horizontal axis shows the fluorescence intensity of iRFP670 (reference), and the vertical axis shows the fluorescence intensity of hmAG1 (reporter).

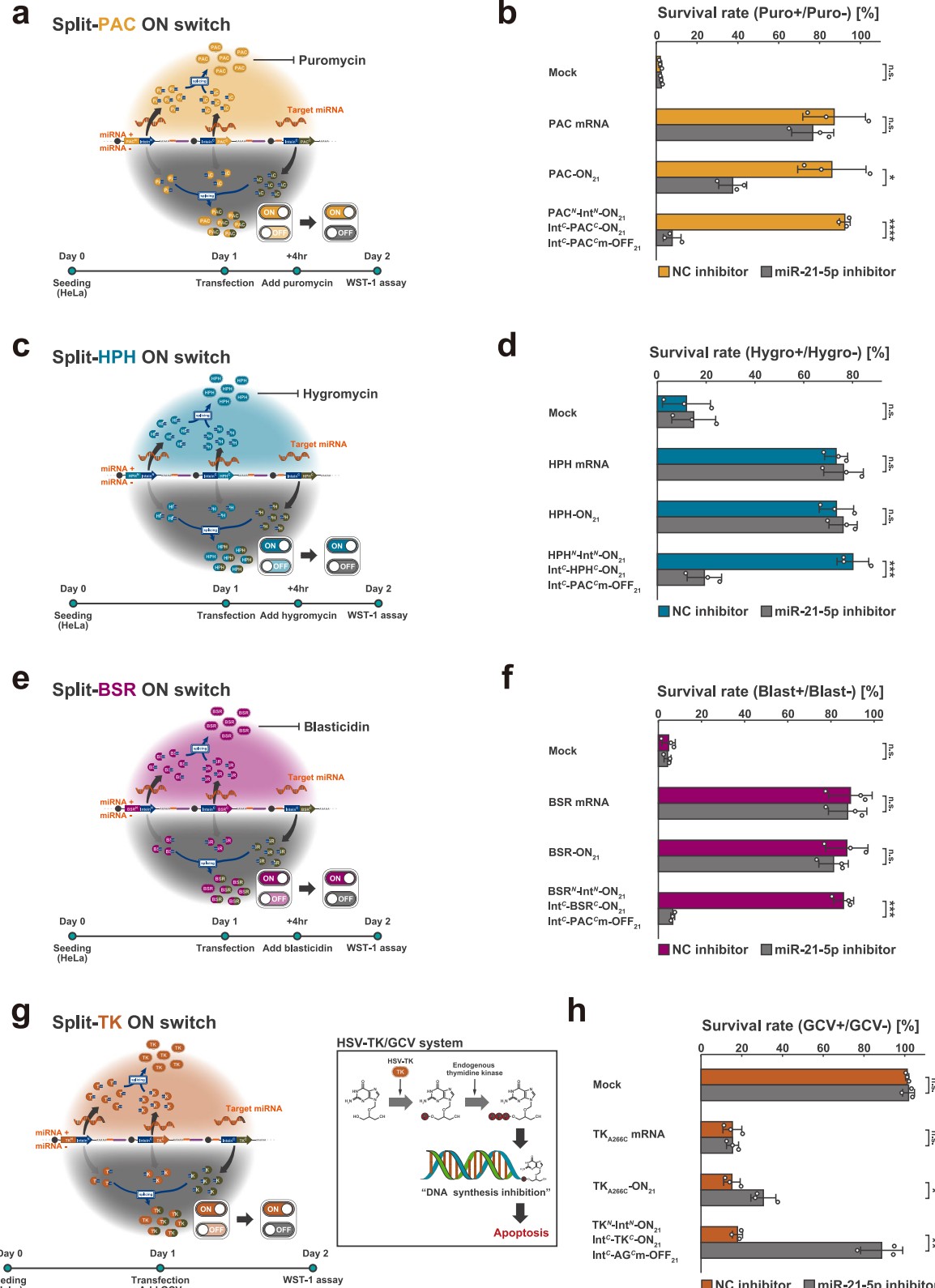

GCV treatment, presumably caused by leaky translation from the ON switch. Meanwhile, the split ON switch pair with a leak-canceller suppressed the leakage of HSV-TK activity, and cell survival remained high in HeLa cells treated with miR-21-5p inhibitor and GCV ($88.7 \pm 10.3\%$).

To confirm the utility of the split ON switch system across cell types and target miRNAs, we also explored the split-PAC ON switch system targeting miR-302a-5p, a human pluripotency marker miRNA

highly active in induced pluripotent stem cells (hiPSCs)[12,49] (Supplementary Fig. 6a). There, we introduced miR-302a-5p responsive switches into hiPSCs (miR-302a-5p-positive cells) and HeLa cells (miR-302a-5p-negative cells). After adding puromycin, we evaluated the viability of these cell types by the WST-1 assay (Supplementary Fig. 6b, c). In hiPSCs, regardless of how the PAC gene was introduced, more than half of the cells survived under conditions with puromycin,

**Fig. 3 | Regulation of various genes by split ON switch system. a, c, e** Schematic illustration of split ON switch system regulating three antibiotic resistance genes: puromycin N-acetyltransferase (PAC) (**a**), hygromycin-B 4-O-kinase (HPH) (**b**), and blasticidin-S deaminase (BSR) (**c**). The experimental time course is shown below each schematic illustration. **b, d, f** The survival rate of HeLa cells treated with miR-21-5p or NC inhibitors as measured by the WST-1 assay. For each condition, the survival rates were calculated by dividing the viability in the presence of antibiotics by that without antibiotics. Error bars represent means ± SD ($n = 3$), and data of each biological replicate are shown as a point. **g** Schematic illustration of split ON switch system regulating herpes simplex virus type 1 thymidine kinase (HSV-TK). **h** The survival rate of HeLa cells treated with miR-21-5p or NC inhibitors as measured by the WST-1 assay. For each condition, the survival rates were calculated by dividing the viability in the presence of ganciclovir (GCV) by that without GCV. Error bars represent means ± SD ($n = 3$), and data of each biological replicate are shown as a point. Statistical analysis by two-sided Welch's $t$test (**b, d, f, h**), *$P < 0.05$, **$P < 0.01$, ***$P < 0.001$, ****$P < 0.0001$, n.s.: not significant ($P > 0.05$). Each $P$-value is listed in Supplementary Table 4. Source data are provided as a Source Data file.

compared to mock conditions without puromycin. A single PAC ON switch rescued HeLa cell survival to a level comparable to PAC mRNA cultured under puromycin, despite the minimal activity of miR-302a-5p. Meanwhile, the split ON switch pair with the leak-canceller significantly suppressed the viability of HeLa cells under puromycin treatment. The suppression became more evident with a larger amount of leak-cancellers. The lower rescue efficiency of hiPSCs compared to HeLa cells, which was observed when treated with puromycin, would reflect the difference in puromycin sensitivity between the cell types. These results indicated the utility of a split switch system across output proteins (fluorescence and antibiotic resistance proteins), cell types (HEK293FT, HeLa, and hiPSCs), and target miRNAs (miR-21-5p and miR-302a-5p) for improving the ON/OFF ratio.

## Cell type-specific selection based on endogenous miRNA activity

Next, we performed a cell type-specific selection experiment based on the differences of endogenous miRNA activity between mixed cell types using split ON switches coding BSR (Fig. 4a). In this experiment, we used HeLa cells (high miR-21-5p activity) as target cells to be selected and HEK293FT cells (low miR-21-5p activity) as non-target cells to be eliminated. To distinguish each cell type from the mixture, we used HeLa cells expressing hmAG1-M9 (HeLa$_{hmAG1-M9}$) and HEK293FT cells expressing iRFP670-M9 (HEK293FT$_{iRFP670-M9}$). After the introduction of the miR-21-5p responsive BSR-ON switch system into a mixture of HeLa and HEK293FT cells, BSR should be active specifically in HeLa cells, resulting only in HeLa cell survival in the presence of blasticidin. After seeding the mixture of HeLa$_{hmAG1-M9}$ and HEK293FT$_{iRFP670-M9}$, we transfected RNA switches and added blasticidin 24 h later, followed by imaging and flow cytometric analysis after three days (Fig. 4b, c, d and Supplementary Fig. 7). When a single ON switch coding BSR was used, the ratio of HeLa to HEK293FT cells (58.6 ± 4.0%) was comparable to the results from untreated cells (no transfection or blasticidin treatment) (58.9 ± 6.6%) and cells with BSR mRNA (60.8 ± 2.2%). Meanwhile, the introduction of the split ON switch pair along with the leak-canceller suppressed HEK293FT cell viability and enabled the purification of HeLa cells to a purity of 96.5 ± 0.61%. From these results, we concluded that the enhanced ON/OFF ratio of the split ON switch system enabled cell type-specific purification based on drug resistance, previously unachievable with a single ON switch.

## Cell type-specific genome editing and application for DMD gene therapy

The versatility in the output genes observed in this methodology can be leveraged for various medical applications, including cell or tissue-specific gene therapy. A major limitation of the CRISPR-Cas9 system in gene therapy is the size of Cas9, which hinders efficient delivery via recombinant adeno-associated viruses (rAAV)[50,51]. To address this issue, optimal split site identification in *Streptococcus pyogenes* Cas9 (*sp*Cas9) and the intein-mediated split-Cas9 system have been demonstrated in previous studies[40,52].

Based on these studies, we designed a split-Cas9 ON switch system and evaluated its ability to enhance the ON/OFF ratio of CRISPR-Cas9 regulation in response to miRNA activity in HeLa cells and hiPSCs (Fig. 5a). We introduced EGFP-targeting sgRNA

(sgRNA$_{EGFP}$) and mRNAs into HeLa cells stably expressing EGFP (HeLa$_{EGFP}$) with either miR-21-5p or NC inhibitors. Cas9 activity under each condition was evaluated based on the EGFP-negative population (Fig. 5b). When a single miR-21-5p-responsive ON switch encoding full-length Cas9 was introduced, high Cas9 activity (EGFP-negative = 71.2 ± 4.8%) was observed even under conditions where miR-21-5p activity was suppressed. In contrast, when miR-21-5p-responsive split-Cas9 ON switches and leak-canceller (a miR-21-5p-responsive OFF switch encoding an inactivated C-terminal fragment of hmAG1) were introduced, Cas9 activity was markedly suppressed (6 × leak canceller: EGFP-negative = 17.5 ± 1.6%) under miR-21-5p-inactvie conditions. Similarly, we introduced sgRNA$_{EGFP}$ and mRNAs into hiPSC stably expressing EGFP (hiPSC$_{EGFP}$)[53,54] with either miR-302a-5p or NC inhibitors and evaluated Cas9 activity (Fig. 5c). When a single miR-302a-5p-responsive ON switch encoding full-length Cas9 was introduced, comparable Cas9 activity was observed under both miR-302a-5p-active (EGFP-negative = 40.7 ± 3.4%) and inactive (EGFP-negative = 39.7 ± 3.4%) conditions. In contrast, when miR-302a-5p-responsive split-Cas9 ON switches and the leak-canceller were introduced, Cas9 activity was markedly suppressed under miR-302a-5p-inactive conditions. Notably, when the leak-canceller was introduced at six times than ON switch, the EGFP-negative population was reduced to 3.0 ± 1.4%, nearly equivalent to that observed in the mock condition (2.6 ± 1.4%). Thus, the split RNA switch system can regulate not only fluorescent and drug-resistance genes but also genome-editing enzymes as output genes to perform cell type-specific genome editing. This system significantly reduces undesired and potentially harmful genome editing in non-target cells.

Furthermore, to demonstrate the practical applicability of cell type-specific Cas9 regulation, we explored its potential application in the treatment of Duchenne muscular dystrophy (DMD). DMD is a severe degenerative muscle disease caused by aberrant splicing of the *dystrophin* gene[55,56]. A promising strategy for a permanent treatment involves introducing the CRISPR-Cas9 system to induce exon skipping in the mutated *dystrophin* gene[57–61]. To develop a safe and practical therapeutic approach, mRNA-based delivery of Cas9 protein has been investigated[62]. However, unintended activation of Cas9 protein in non-target cells and tissues in vivo poses a significant risk of off-target genome editing, leading to potentially harmful mutations[63–65]. Therefore, cell type-specific activation of Cas9 is crucial for ensuring the safety of CRISPR-Cas9-mediated DMD therapy.

The deletion of exon 44 in the *dystrophin* gene is the third or fourth most common deletion in patients with DMD[66–68]. As potential therapeutic strategies for exon 44-deleted DMD patients, three approaches have been proposed: (i) disrupting the splicing acceptor of exon 45 to connect exons 43 and 46, thereby restoring the reading frame; (ii) inducing a frameshift by introducing small indels; and (iii) inserting exon 44 in front of exon 45. The effective implementation of these strategies requires the induction of a double-strand break at the splicing acceptor site immediately upstream of exon 45 (DMDex45). Therefore, we sought to demonstrate the cell type-specific disruption of the DMDex45 splicing acceptor site based on differential miRNA activity, effectively suppressing undesired Cas9 activity in non-target cells. In this study, we evaluated DMDex45 editing efficiency using a

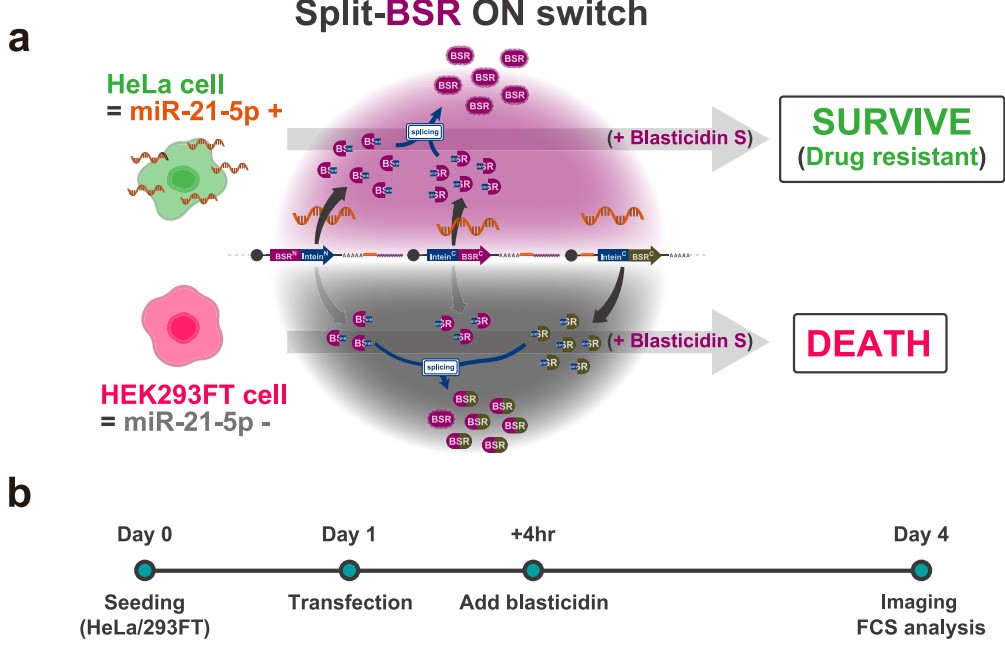

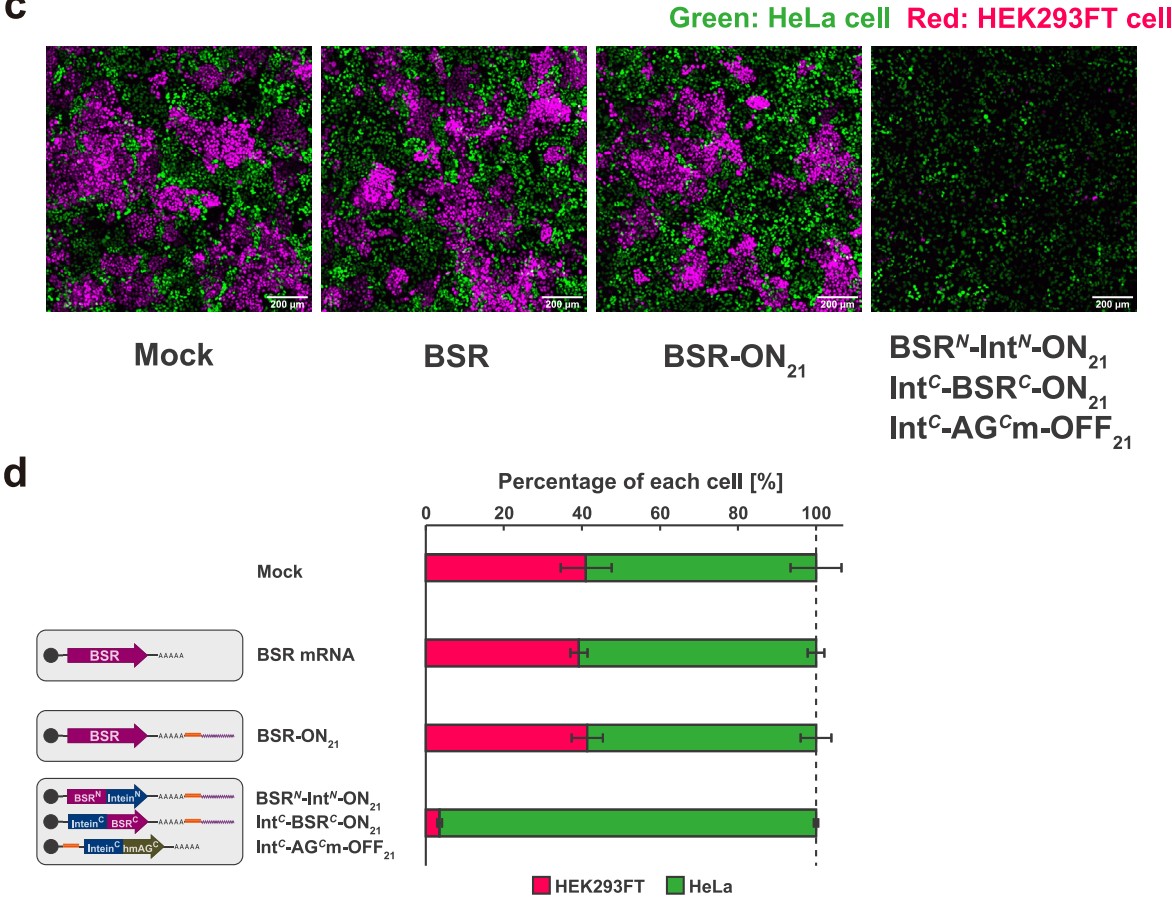

**Fig. 4 | Cell type-specific selection based on endogenous miRNA activity.**
**a** Schematic illustration of the miR-21-5p-responsive split ON switch system regulating blasticidin-S deaminase (BSR). This system suppresses the leaky activity of BSR in miR-21-5p negative cells (HEK293FT cells) compared to the conventional single ON switch while maintaining the drug-resistance gene in miR-21-5p positive cells (HeLa cells). **b** Schematic illustration of the experimental procedure. **c** Representative merged fluorescence images of the mixture of HeLa and HEK293FT cells transfected with mRNAs. HeLa cells stably expressing hmAG1-M9 (HeLa_hmAG1-M9) and HEK293FT cells stably expressing iRFP670-M9 (HEK293FT_iRFP670-M9) were used to distinguish each cell line. Scale bar, 200 μm. **d** Percentage of HeLa and HEK293FT cells. The number of each cell type was counted inside the squares in Supplementary Fig. 7, and the percentage of each cell line was calculated. Error bars represent means ± SD ($n = 3$, biological replicates). Source data are provided as a Source Data file.

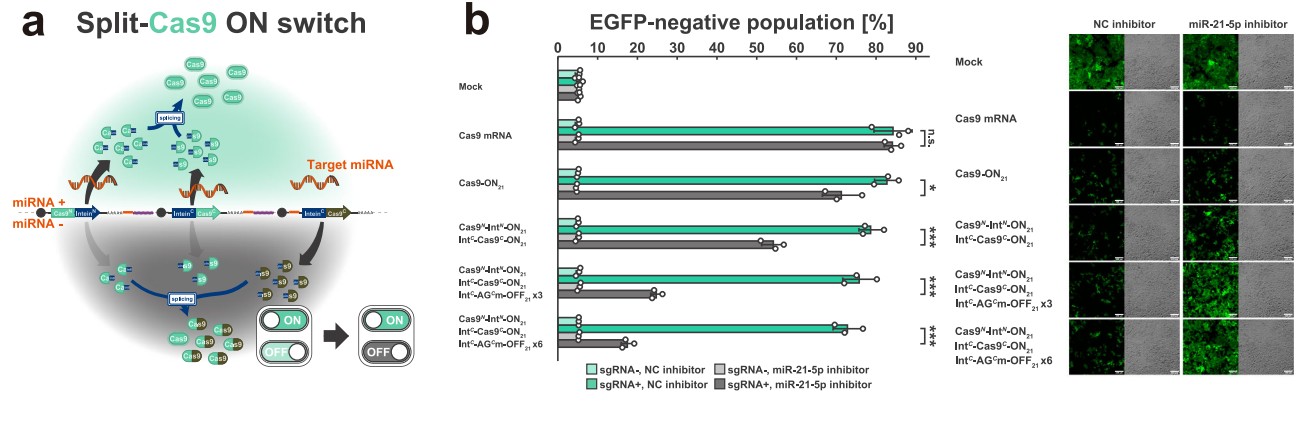

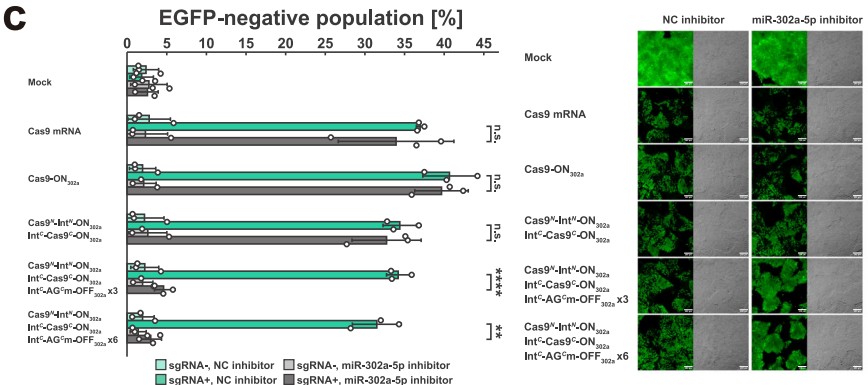

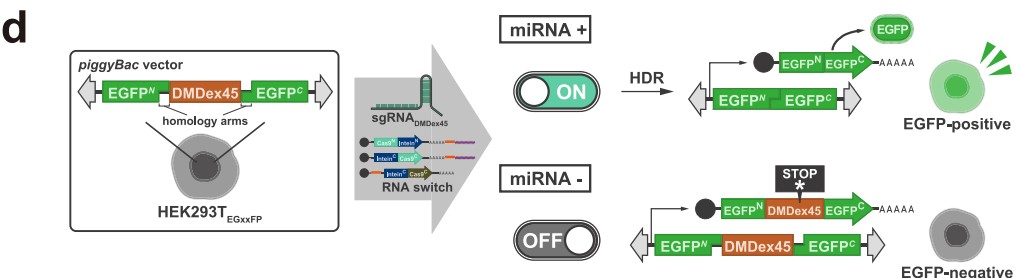

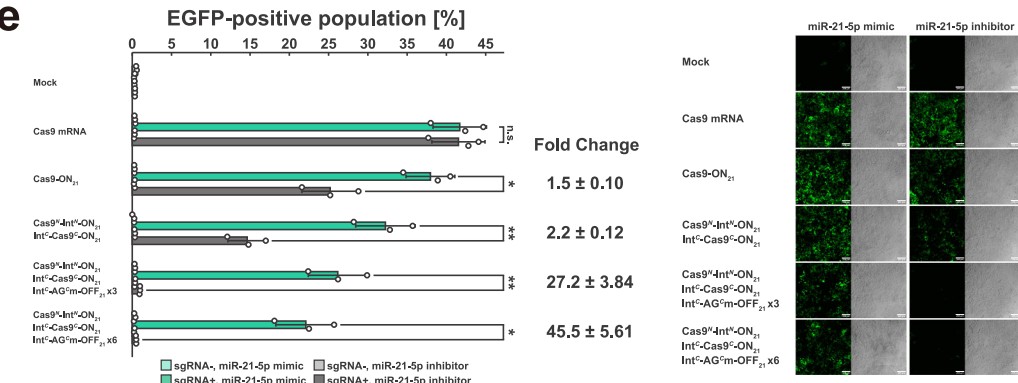

reporter HEK293T cell (HEK293T$_{EGxxFP}$)[57,69], which exhibits EGFP fluorescence upon genome editing of DMDex45 splicing acceptor (Fig. 5d). We introduced DMDex45-targeting sgRNA (sgRNA$_{DMDex45}$) and mRNAs into HEK293T$_{EGxxFP}$ treated with either a miR-21-5p mimic or inhibitor, and evaluated the disruption of the DMDex45 splicing acceptor based on the EGFP-positive population (Fig. 5e). When a single miR-21-5p-responsive ON switch encoding full-length Cas9 was introduced,

undesirable Cas9 activity leakage was observed (EGFP-positive = 25.2 ± 3.6%) even when miR-21-5p activity was suppressed. Consequently, the fold change in DMDex45 splicing acceptor disruption events between miR-21-5p-active and -inactive conditions was limited to 1.5-fold. In contrast, when miR-21-5p-responsive split-Cas9 ON switches and six times the amount of leak-canceller were introduced, Cas9 activity was markedly suppressed (EGFP-positive = 0.5 ± 0.02%) under

**Fig. 5 | Cell type-specific genome editing and application for DMD gene therapy. a** Schematic illustration of the split ON switch system regulating *sp*Cas9. **b** EGFP-negative population of HeLa$_{EGFP}$ treated with miR-21-5p or NC inhibitors in the presence or absence of sgRNA$_{EGFP}$. Error bars represent means ± SD ($n = 3$), and data of each biological replicate are shown as a point. Statistical analysis by two-sided Welch's *t*test, *$P < 0.05$, ***$P < 0.001$, n.s.: not significant ($P > 0.05$). Each *P*-value is listed in Supplementary Table 4. Source data are provided as a Source Data file. Green fluorescence (left) and bright-field (right) images are shown for each condition. Scale bar, 200 μm. **c** EGFP-negative population of hiPSC$_{EGFP}$ treated with miR-302a-5p or NC inhibitors in the presence or absence of sgRNA$_{EGFP}$. Error bars represent means ± SD ($n = 3$), and data of each biological replicate are shown as a point. Statistical analysis by two-sided Welch's *t*test, **$P < 0.01$, ****$P < 0.0001$, n.s.: not significant ($P > 0.05$). Each *P*-value is listed in

Supplementary Table 4. Source data are provided as a Source Data file. Green fluorescence (left) and bright-field (right) images are shown for each condition. Scale bar, 200 μm. **d** Schematic illustration of dystrophin exon 45 (DMDex45) editing strategy. In this study, we evaluated DMDex45 editing efficiency using a reporter HEK293T cell (HEK293T$_{EGxxFP}$)[57,62], which exhibits EGFP fluorescence upon genome editing of DMDex45 splicing acceptor. **e** EGFP-positive population of HEK293T$_{EGxxFP}$ treated with miR-21-5p inhibitor or miR-21-5p mimic in the presence or absence of sgRNA$_{DMDex45}$. Error bars represent means ± SD ($n = 3$), and data of each biological replicate are shown as a point. Statistical analysis by two-sided Welch's *t*test, *$P < 0.05$, **$P < 0.01$, n.s.: not significant ($P > 0.05$). Each *P*-value is listed in Supplementary Table 4. Source data are provided as a Source Data file. Green fluorescence (left) and bright-field (right) images are shown for each condition. Scale bar, 200 μm.

miR-21-5p-inactive conditions, although a slight reduction was also observed under miR-21-5p-active conditions. Notably, when the leak-canceller was introduced at three- or six-fold excess relative to the ON switch, the fold change in DMDex45 splicing acceptor disruption events between miR-21-5p-active and -inactive conditions dramatically increased to 27.2 (± 3.8) -fold and 45.5 (± 5.6) -fold, respectively. These results support the feasibility of using the split RNA switch for accurate, cell type-specific regulation of Cas9 activity to minimise off-target effects and enhance the safety of CRISPR-Cas9-mediated DMD therapy.

## Improving the performance of two-outputs using a split toggle-like system

In flow cytometry (FC) analysis, miRNA-responsive RNA switches can be a powerful tool to distinguish different cell types based on their endogenous miRNA profiles[5,12,15]. However, a single RNA switch often fails to completely separate two cell populations, mainly due to a low ON/OFF ratio of RNA switch and minute differences in endogenous miRNA activity[6,15]. In contrast, a "toggle-like system," a two-output system switching its outputs in response to the target miRNA activity, offers clearer separation in FC 2D plots (Supplementary Fig. 8a). A simple way to construct a toggle-like system using RNA switches is to introduce a pair of ON and OFF switches, each coding different fluorescent proteins, targeting the same miRNA ("normal toggle-like system") (Fig. 6a, left). Nevertheless, in the case of the normal toggle-like system, the leaky translation from each switch could still undermine the binariness of the system.

We hypothesised that the combination of protein splicing and RNA switch could enhance the binariness of a toggle-like system. As a proof-of-concept, we devised the "split toggle-like system" using two pairs of switches: two ON switches coding the N-terminal and C-terminal of hmAG1 and two OFF switches coding the N-terminal and C-terminal of iRFP670 (Fig. 6a, right). In miRNA + cells, the high abundance of hmAG1 fragments translated from ON switches inhibits splicing between iRFP670 fragments—leaked from the OFF switches—thereby inhibiting the formation of functional iRFP670. In miRNA-cells, the opposite is expected: abundant iRFP670 fragments translated from OFF switches inhibit splicing between hmAG1 fragments leaked from the ON switches, thus inhibiting the formation of functional hmAG1. In the entire system, the binary trend, in which miRNA+ cells exhibit green fluorescence and miRNA- cells display red fluorescence, should be intensified, resulting in better separation of the two cell populations in FC 2D plots (Supplementary Fig. 8b, c). To test the system, we transfected the four switches (AG$^N$-Int$^N$-ON$_{21}$, Int$^C$-AG$^C$-ON$_{21}$, RF$^N$-Int$^N$-OFF$_{21}$, Int$^C$-RF$^C$-OFF$_{21}$) into HEK293FT cells with varying concentrations of miR-21-5p mimic, followed by flow cytometric analysis 24 h later (Fig. 6b, c and Supplementary Fig. 9). In the split toggle-like system, hmAG1 fluorescence intensity under the condition without miR-21-5p mimic was suppressed to 8.4 ± 1.6% of the condition with 2 nM mimic, significantly lower than that in the normal toggle-like system (32.4 ± 3.4%). Notably, iRFP670 fluorescence intensity at sufficient mimic concentrations (2 nM) was also suppressed to 2.1 ± 0.35%

of the condition without mimic and significantly lower than that in the normal toggle-like system (6.9 ± 1.5%). Additionally, the distance between the centroids of the two cell populations at 0 nM mimic and each population in logarithmic 2D plots was approximately 1.6–2.3 times greater compared to the normal toggle-like system (Fig. 6d). From these results, we concluded that splitting the output genes coded in RNA switches in a toggle-like system improves the distinction of the two states, which allows for clearer classification of cell types based on differential miRNA activity.

## Construction of mRNA-based intracellular two-input logic gates

In addition to improving the ON/OFF ratio of the ON switch system and the binariness of the toggle-like system, the combination of RNA switch and protein splicing enables the construction of multi-input intracellular systems that sense multiple miRNAs to regulate output protein function with RNA-only delivery. For example, introducing a pair of ON switches with different miRNA target sites, each of which codes the N- and C-terminal fragments of a gene, allows for the construction of an "AND gate," in which the protein of interest functions only in cells that exhibit both miRNA activities (Fig. 7a).

As a proof-of-concept experiment, we constructed four types of two-input logic gates (NOR, A AND NOT B, NOT A AND B, and AND) in HEK293FT cells treated with different combinations of miR-21-5p or miR-302a-5p mimics. After introducing ON or OFF switches responsive to miR-21-5p or miR-302a-5p, we evaluated their performance in a reporter assay using hmAG1 as the output (Fig. 7b and Supplementary Fig. 10). Each system is expected to exhibit hmAG1 fluorescence activity only in cells with specific combinations of miR-21-5p and miR-302a-5p activities (denoted as [00],[10],[01],[11]). In the three systems (A AND NOT B, NOT A AND B, and AND) in which we used ON switches, we added a leak-canceller, an OFF switch coding an inactive, opposing fragment with the same miRNA target site as the corresponding ON switch, to prevent translational leakage from ON switches. In the reporter assay, the values of the OFF state were suppressed to less than 10% of the ON state in the NOR gate and less than 20% of the ON state in the A AND NOT B, NOT A AND B, and AND gates. FC 2D plots indicated that all the logic gates regulated hmAG1 dramatically enough to distinguish a cell subpopulation from other states to near completion in the scatter plots (Fig. 7b). Considering the mechanisms of each logic gate shown in Supplementary Fig. 10, increasing the amount of leak-canceller can further suppress the leakage in OFF states. These results show that the integration of translational control by multiple RNA switches via protein splicing simultaneously enables the construction of RNA-based intracellular two-input systems in addition to the suppression of leakage in their OFF states.

## Construction of two-type-input logic circuit for simultaneous detection of protein and miRNA

The range of potential targets for split RNA switches is not limited to miRNA but could be readily extended to other types of biomolecules, such as proteins, by utilising protein-responsive switches[70]. It is

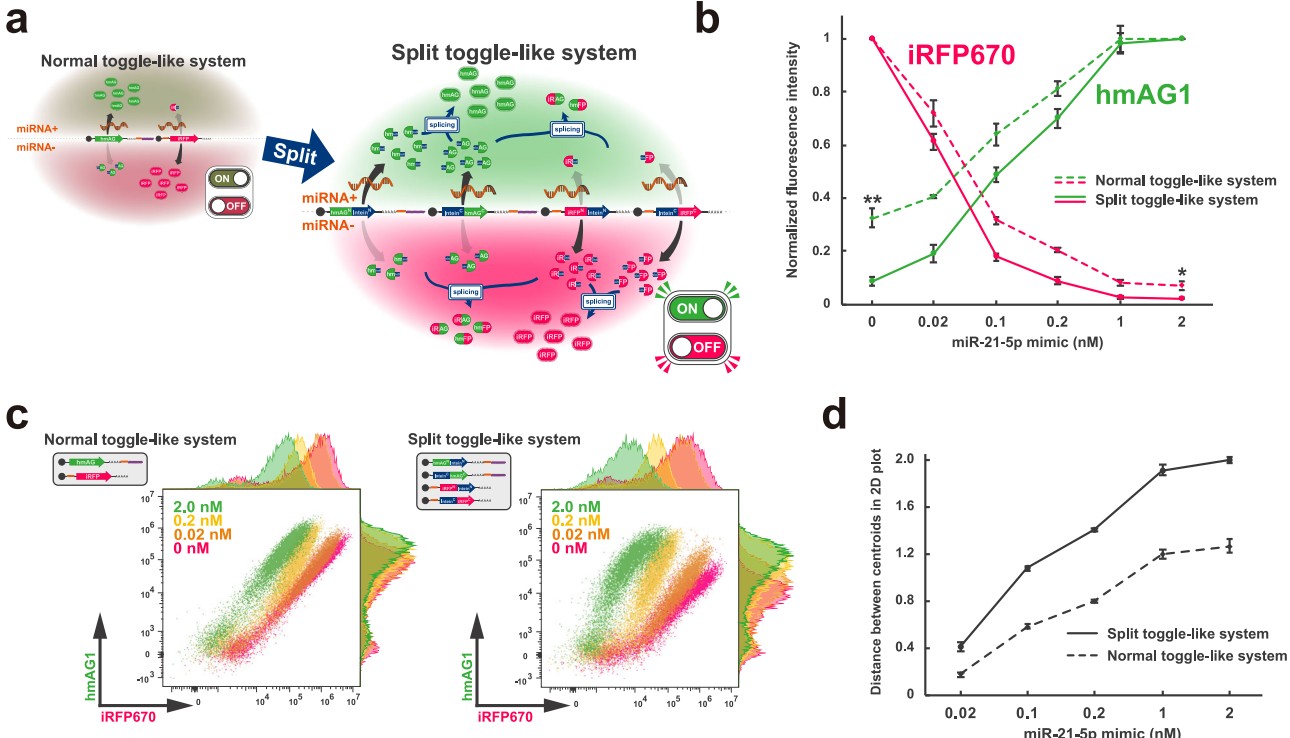

**Fig. 6 | Engineering of a toggle-like system using split-intein. a** Schematic illustration of a toggle-like (two-output) system that displays hmAG1 fluorescence in the presence of target miRNA activity and iRFP670 fluorescence in its absence. "Normal toggle-like system" refers to a system composed of an ON switch coding full-length hmAG1 and an OFF switch coding full-length iRFP670, and "split toggle-like system" refers to a system composed of a pair of ON switches coding split hmAG1 fragments and a pair of OFF switches coding split iRFP670 fragments. **b** Normalised hmAG1 and iRFP670 fluorescence intensity of HEK293FT cells treated with various miR-21-5p mimic concentrations. Normalised hmAG1 intensity was calculated by normalising the intensity at 2 nM mimic concentration, and normalised iRFP670 intensity was calculated by normalising the intensity at 0 nM mimic

concentration. Error bars represent means ± SD ($n = 3$, biological replicate). Statistical analysis by two-sided Welch's $t$test, *$P < 0.05$, **$P < 0.01$. Each $P$-value is listed in Supplementary Table 4. Source data are provided as a Source Data file. **c** Representative 2D flow cytometry plots of HEK293FT cells treated with various miR-21-5p mimic concentrations. The horizontal axis shows the fluorescence intensity of iRFP670 (reference), and the vertical axis shows the fluorescence intensity of hmAG1. **d** Euclidean distance between the centroid of the 0 nM mimic concentration plot and those of various mimic concentrations in the logarithmic 2D plot in (**c**). Error bars represent means ± SD ($n = 3$, biological replicate). Source data are provided as a Source Data file.

possible to construct intracellular logic gates that regulate gene expression in response to combinations of different types of biomolecules. For example, the introduction of both a protein- and a miRNA-responsive OFF switch, each encoding the N- and C-terminal fragments of a gene, enables the construction of a protein/miRNA-responsive NOR gate. In this system, the protein of interest (output) is functional only in cells lacking both the target protein and miRNA (Fig. 8a).

To experimentally validate the concept of a protein/miRNA-responsive logic gate, we implemented a "L7Ae/miR-21 NOR" gate composed of an L7Ae-responsive OFF switch[16,71] and a miR-21-5p-responsive OFF switch[5], with hmAG1 as the output (Fig. 8b, upper). A reporter assay was performed in HEK293FT cells transfected with L7Ae-encoding mRNA, miR-21-5p mimic, or both, to evaluate the performance. In the assay, the value of the OFF state was suppressed to less than 10% of the ON state in the NOR gate. FC 2D plots indicated that the NOR gate regulated hmAG1 expression sufficiently to produce a clear cell separation. Similarly, using the same concept, we designed a "LIN28A/miR-302a-NOR" gate to detect the combination of LIN28A and miR-302a-5p. LIN28A is a marker protein that binds to a specific class of RNAs and is highly expressed in human pluripotent stem cells[72–74]. To construct the logic gate, we employed a LIN28A-responsive OFF switch[8] and a miR-302a-5p-responsive OFF switch[12] (Fig. 8b, lower). We evaluated the performance of this gate by transfecting HEK293FT cells with LIN28A-encoding mRNA, miR-302a-5p mimic, or both. As a result, the values of the OFF state were suppressed to less than 15% of the ON state in the NOR gate. Again, we observed a clear cell separation on the FC 2D plot, suggesting that the NOR gate

functioned as expected to regulate hmAG1 expression to enable purification of target cells from other states.

Because both LIN28A and miR-302a-5p are pluripotency markers, the LIN28A/miR-302a-NOR gate could function as an mRNA-based synthetic circuit to suppress protein activity selectively in hiPSCs. To test the hypothesis, we introduced a miR-302a-responsive OFF switch, a LIN28A-responsive OFF switch, and the LIN28A/miR-302a NOR gate into HEK293FT cells and hiPSCs, and compared the fluorescence activity of the output reporter, hmAG1, between these cell types (Fig. 8c). The fold change in fluorescence between HEK293FT and hiPSCs was 13.0 ± 3.7 for the miR-302a-responsive OFF switch and 3.23 ± 0.3 for the LIN28A-responsive OFF switch. In contrast, the LIN28A/miR-302a NOR gate exhibited a fold change of 27.2 ± 3.9, representing an over twofold increase compared to the miR-302a-responsive OFF switch. Thus, the split RNA switch system enables the integration of outputs from existing mRNA-based gene regulation technologies solely by designing the ORF sequence and facilitates the simultaneous detection of multiple types of biomolecules. We demonstrate that mRNA-based multi-type-input synthetic circuits hold great promise for achieving robust gene regulation based on endogenous cellular information.

### Construction of three-input logic circuits using two orthogonal split intensities

In addition to the NpuDnaE split intein used in this study, there are different types of inteins with splicing-orthogonality to each other[75]. With split gene fragments from multiple orthogonal inteins, we design

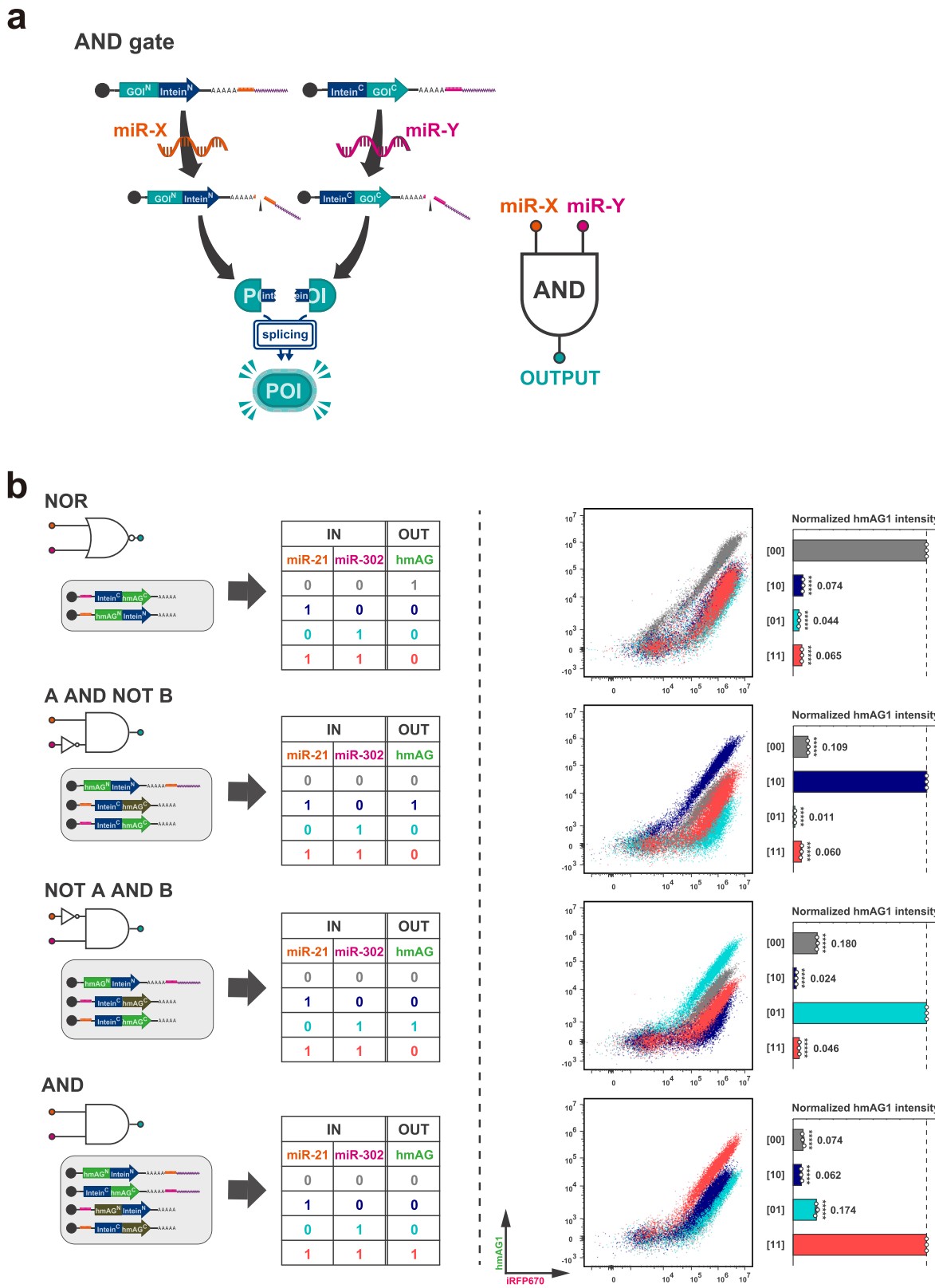

multi-input systems to regulate gene expression in response to more than three types of miRNAs. For example, a set of three OFF switches with different miRNA target sites, each of which codes the tripartite fragments of a gene, allows for the construction of a three-input "NOR gate," that produces a functional output protein only in the absence of all target miRNAs (Fig. 8d, left). Similarly, introducing a set of three ON switches with different miRNA target sites, each of which codes the

tripartite fragments of a gene, allows for the construction of a three-input "AND gate," that produces a functional output protein only in the presence of all target miRNAs (Fig. 8d, right).

Here, we aimed to demonstrate three-input systems using hmAG1 as output. To construct such three-input systems, in which we induce the reassembly of hmAG1 tripartite fragments with two simultaneous orthogonal protein splicing reactions, we chose the SspDna$_{\text{BM86}}$ mini-

**Fig. 7 | Intracellular two-input logic circuit using split-intein. a** Schematic illustration of a two-input logic gate using split-intein, exemplifying an AND gate, in which an output protein is activated only in the presence of both of two different target miRNAs. **b** Four types of two-input logic circuits. Different colours correspond to those in the truth table, representative 2D flow cytometry plots, and bar graphs, respectively. MiR-21-5p and miR-302a-5p mimics were used as inputs. For example, the input pattern [10] coloured in blue means miR-21-5p was present while miR-302a-5p was absent. The representative 2D flow cytometry plot for each circuit is shown as an overlay of scatter plots for all four input patterns. The relative fluorescence intensity of hmAG1 (hmAG1/iRFP670) was normalised by the highest value for each circuit ([00] in NOR,[10] in A AND NOT B,[01] in NOT A AND B, and [11] in AND). Error bars represent means ± SD (*n* = 3), and data of each biological replicate are shown as a point. Statistical analysis by two-sided Dunnett's test for the ON state of each circuit, *****$P < 0.00001$. Each *P*-value is listed in Supplementary Table 4. Source data are provided as a Source Data file.

intein[76], a variant of the wild-type SspDna[77] with eight amino acid substitutions that enhance splicing efficiency, as the "second split-intein," given its known splicing-orthogonality to Npu-DnaE, the "first split-intein" used in this research[37]. In the case of the SspDnaB$_{BM86}$ mini-intein, efficient protein splicing requires a glycine at the C-terminal position of the N-terminal side extein[78]. To satisfy this requirement, we picked up three candidates "second split site" in hmAG1—130 G/131 P, 152 G/153 V, and 166 G/167 G—in addition to the "first split site" (62 F/63Q). Each site is located in the surface loop of hmAG1 and contains a glycine residue on the N-terminal side. Furthermore, to align the N-terminal position of the C-terminal side extein with the "native context" of SspDnaB$_{BM86}$ mini-intein, we substituted each C-terminal side amino acid residue of the split sites with serine (130 G/131S, 152 G/153S, and 166 G/167S)[78]. For the SspDnaB$_{M86}$ mini-intein, several potent split sites have been identified in previous studies[79]. In this study, we selected and evaluated two commonly used intein split sites: 11 A/12S ("type-a") and 105Q/106 L ("type-b"). To validate all six possible combinations (three candidate split sites in hmAG1 and two in SspDnaB), we designed a total of 12 fragmented variants and performed reporter assay after introducing either the N-terminal or C-terminal fragments alone, or both fragments together, in HEK293FT cells (Supplementary Fig. 11a). As a result, no fluorescence was observed when either the N-terminal or C-terminal fragment was introduced alone, regardless of the split pattern (Supplementary Fig. 11b). The strongest fluorescence signal was detected when both fragments were introduced in the "130 G/131 P × b-type" combination. Therefore, we selected this combination as the "second split site" for the subsequent construction of three-input logic circuits.

To build the miR-21/302a/206 NOR gate, we designed three types of split RNA switches—miR-21-5p-, miR-302a-5p-, and miR-206-responsive OFF switches—each encoding one of the hmAG1 tripartite fragments: AG$^N$ (1–62)-Npu$^N$, Npu$^C$-AG$^{MID}$ (63–130)-Ssp$^N$, and Ssp$^C$-AG$^C$ (131–226). These split RNA switches were introduced into HEK293FT cells treated with different combinations of miR-21-5p, miR-302a-5p, and miR-206 mimics, followed by a reporter assay. As a result, hmAG1 fluorescence was observed in cells with neither miR-21-5p, miR-302a-5p, nor miR-206 activities (denoted as [000] in the histogram), with an intensity more than 16 times higher than that observed in cells under other conditions (Fig. 8e, left). Next, to implement the miR-21/302a/206 AND gate, we designed six types of split RNA switches: miR-21-5p-, miR-302a-5p-, and miR-206-responsive ON switches, each encoding one of the hmAG1 tripartite fragments, along with their corresponding leak cancellers (miR-21-5p-, miR-302a-5p-, and miR-206-responsive OFF switches encoding inactivated middle fragments with a G65A mutation (i.e., AG$^{MID}$m (63–130)-Npu$^N$, Npu$^C$-AG$^{MID}$m-Ssp$^N$, and Ssp$^C$-AG$^{MID}$m)). These six split RNA switches were introduced into HEK293FT cells treated with different combinations of miR-21-5p, miR-302a-5p, and miR-206 mimics, followed by a reporter assay. As a result, hmAG1 fluorescence was observed predominantly in cells with all of miR-21-5p, miR-302a-5p, and miR-206 activities (denoted as [111] in the histogram), with an intensity more than 5 times higher than in cells under other conditions (Fig. 8e, right).

## Discussion

RNA switch is an RNA-based translational control tool that holds great potential for regenerative medicine and mRNA therapeutics to enable cell type-specific gene regulation without the risk of harmful genomic mutagenesis. While prior studies have shown the impact of RNA switch, small ON/OFF ratios of output and difficulties in finding optimal target candidate molecules have often prevented practical applications such as cell classification and purification. In this study, to address these issues, we have designed "split RNA switches" that leverage protein splicing, a type of post-translational modification. The combination of RNA switch and protein splicing allows the integration of translational outputs from multiple RNA switches at the post-translational level, thereby achieving high cell type specificity. We designed the split RNA switches, RNA switches coding split genes conjugated by split-intein sequences (Fig. 1). By using a set of split RNA switches coding functional or non-functional protein fragments, we implemented a miRNA-responsive ON switch system in HEK293FT cells with over 25-fold ON/OFF ratio, by suppressing undesired leaky OFF-state output (Fig. 2). Results from the hmAG1 reporter assay revealed that optimising the amount of OFF switch could enable further suppression of the leakage in the OFF state. Although we used inactive C-terminal fragments with a single amino acid mutation as the output protein for the leak-canceller in this research, it is unnecessary to use mutated C-terminal fragments for each gene because mutated C-terminal fragments for the leak-canceller are only required to form an inactive protein through splicing with the N-terminal fragment. Considering this point, the extein of the leak-canceller should work with a shorter length, and thus smaller volumes of leak-canceller mRNA could enable the same level of leak-cancelling efficiency as in this study.

We demonstrate efficient cell type-specific cell fate control with split ON switch systems outputting three types of antibiotic-resistant genes (Fig. 3). In these experiments, we successfully reduced OFF-target cell survival. In addition, we confirmed that the split-BSR ON switch system with a leak-canceller allowed for the efficient purification of HeLa cells from a mixture with HEK293FT cells based on the difference of endogenous miRNA activity (Fig. 4). In the mock condition of the cell purification experiment, the final proportion of HEK293FT cells had increased to more than 40% of the total after five days, despite an initial seeding at a HeLa$_{hmAG1-M9}$:293FT$_{iRFP670-M9}$ ratio of 5:1. This result indicates that HEK293FT cells are less sensitive to blasticidin-dependent apoptotic pathways and more resistant to cell death than HeLa cells. The split BSR-ON switch system with leak-canceller achieved a high ON/OFF ratio, sufficient to purify HeLa cells beyond the difference of blasticidin-sensitivity of cell types. Furthermore, we demonstrate the activation of HSV-TK in a manner dependent on the target miRNA and GCV (Fig. 3) to explore the application of the split-ON switch system in medical scenarios, inducing cell type-specific apoptosis and eliminating unwanted cells without direct off-target effects in the absence of GCV. These results indicate that split RNA switches can provide precise cell fate control, which plays a pivotal role in cell therapy and regenerative medicine, by simply introducing mRNAs. Moreover, we applied the split ON switch system to the CRISPR-Cas9 platform in human cancer cells and pluripotent stem cells, enabling miRNA-dependent gene knockout with minimal off-target effects (Fig. 5). We also achieved miRNA-dependent editing of DMD exon 45 with a high ON/OFF ratio (> 45-fold differences between miRNA +/− cells), demonstrating a promising strategy for DMD gene therapy. The split RNA switch represents a powerful

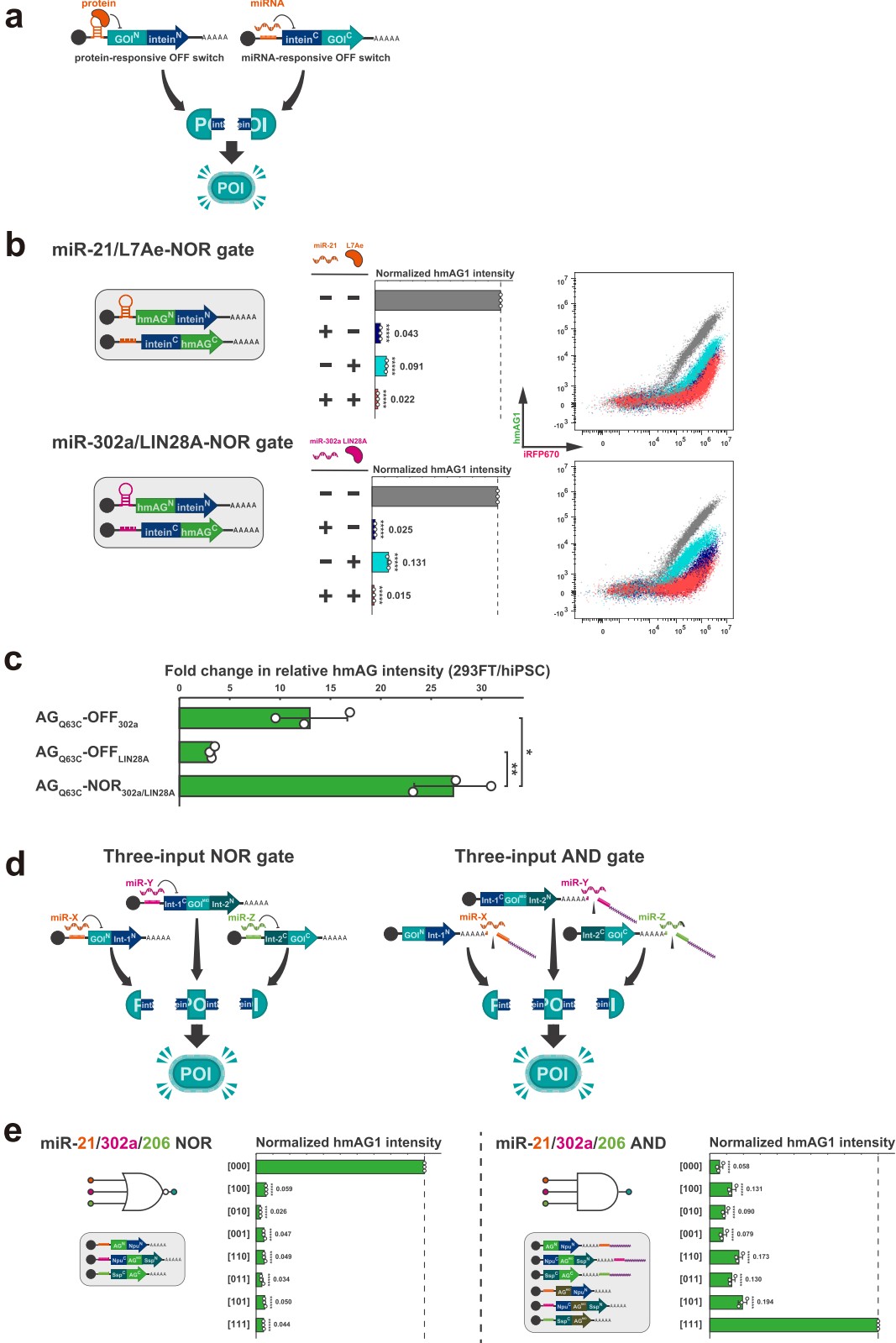

approach for mRNA-based gene therapy, effectively minimising unintended Cas9 activity in non-target cells.

Although the mechanism of the miRNA-responsive ON switch is hypothesised to involve the ability of the extra sequence added to the 3' end to suppress gene expression in the absence of miRNA, and a perfect complementary match by the endogenous target miRNA to trigger the enzymatic removal of the extra sequence by the Argonaute

2 protein[31], thereby relieving the suppression, the detailed mechanism remains unclear[6]. Therefore, a rational design strategy for the extra sequence to achieve improved ON/OFF control has not yet been established. Even though translational control technologies such as the miRNA-responsive ON switch, whose mechanism and molecular design strategy remain poorly understood and thus continue to be hindered by low ON/OFF ratios, we have illustrated significant

**Fig. 8 | Construction of two-type-input and three-input logic circuit.**
**a** Schematic illustration of a two-type-input logic gate using split-intein, exempli-
fying a protein/miRNA-resonsive NOR gate, in which an output protein is activated
only in the absence of both target protein and miRNA. **b** Two types of two-type-
input logic circuits. L7Ae/miR-21-NOR gate exhibits hmAG1 fluorescence only in the
absence of both L7Ae and miR-21-5p (upper). LIN28A/miR-302a-NOR gate exhibits
hmAG1 fluorescence only in the absence of both LIN28A and miR-302a-5p (lower).
Colours correspond to those in the bar graphs and representative 2D flow cyto-
metry plots, respectively. Error bars represent means ± SD ($n = 3$), and data of each
biological replicate are shown as a point. Statistical analysis by two-sided Dunnett's
test for the ON state of each circuit, *****$P < 0.00001$. Each $P$-value is listed in
Supplementary Table 4. Source data are provided as a Source Data file. **c** Compar-
ison of fold changes in cell type-dependent protein activity regulation among miR-
302a-responsive OFF switch, LIN28A-responsive OFF switch, and LIN28A/miR-302a
NOR gate. The fold changes in relative hmAG1 intensity were calculated as hmAG1/

iRFP670 of HEK293FT cells divided by that of hiPSC. Error bars represent
means ± SD ($n = 3$), and data of each biological replicate are shown as a point.
Statistical analysis by two-sided Welch's $t$test, *$P < 0.05$, **$P < 0.01$. Each $P$-value is
listed in Supplementary Table 4. Source data are provided as a Source Data file.
**d** Schematic illustration of a three-input logic gate using two orthogonal split-
intein, exemplifying a three-input NOR gate, in which an output protein is activated
only in the absence of all target miRNAs, and a three-input AND gate, in which an
output protein is activated only in the presence of all target miRNAs. **e** Validation of
miR-21/302a/206 NOR gate (left) and miR-21/302a/206 AND gate (right) in
HEK293FT cell treated with different combinations of miR-21-5p, miR-302-5p, miR-
206 mimics. Error bars represent means ± SD ($n = 3$), and data of each biological
replicate are shown as a point. Statistical analysis by two-sided Dunnett's test for
the ON state of each circuit, *****$P < 0.00001$. Each $P$-value is listed in Supplemen-
tary Table 4. Source data are provided as a Source Data file.

improvements by incorporating well-studied post-translational con-
trol elements. As a result, our split RNA switch system improves the
ON/OFF ratio to a level for practical use, as demonstrated in cell type-
specific purification, genome editing, and DMD gene therapy
(Figs. 3–5).

In addition to the "single-input single-output" systems, we
designed a "single-input two-output system" in which each of the two
types of fluorescent proteins cancels the fluorescence of the other. The
split toggle-like system separated cell subpopulations with different
miRNA levels more clearly than using pairs of ON/OFF switches coding
full-length proteins (Fig. 6). The leak-canceller used in this experiment,
coding an inactive C-terminal fragment, resulted in inactive cancelling
products by splicing with cognate N-terminal fragments. By contrast,
engineering the cancelling products to have functional properties
could be an alternative strategy for implementing two-output systems.
For example, consider the blue fluorescent protein (BFP) and green
fluorescent protein (GFP), which differ by a single amino acid[80]. The
introduction of normal mRNA for the N-terminal fragment of GFP,
along with an ON switch for the GFP C-terminal fragment and an OFF
switch for the BFP C-terminal fragment, could result in green fluores-
cence in miRNA + cells and blue fluorescence in miRNA- cells, with the
green fluorescence effectively quenched. A more sensitive two-output
system could identify and isolate a specific cell type based on the
miRNA profile from highly heterogeneous cell populations such as
hematopoietic lineages, in which miRNAs act as master regulators of
transcriptional programs, and their expression patterns are known to
reflect biological relationships among them[22,81].

Moreover, we demonstrate four types of intracellular two-input
logic gates with split RNA switches (Fig. 7). We confirmed that these
two-input logic gates output high protein activity in cells with a specific
combination of two miRNA activities. In addition, we extended this
concept by integrating translation control mediated by both a protein-
and a miRNA-responsive RNA switch, thereby constructing an mRNA-
based logic circuit capable of simultaneously detecting proteins and
miRNAs (Fig. 8a–c). By designing the LIN28A/miR-302a NOR gate that
detects both an endogenously expressed protein and a miRNA specific
to hiPSCs, we achieved an ON/OFF ratio more than twice that of the
previously reported miR-302a-5p-responsive OFF switch[12]. The miR-
302a-5p-responsive OFF switch could be applied in regenerative
medicine, for example, to enable the purification of differentiated
midbrain dopaminergic (mDA) neurons from the mixture with residual
hiPSCs, thus eliminating the need for a cell sorter[12]. The LIN28A/miR-
302a NOR gate introduced in this study could further enhance cell type
specificity and purification efficiency in such applications. Moreover,
we successfully constructed a more complex three-input logic circuit
in human cells using two types of orthogonal split inteins (Fig. 8d, e).
The development of such intricate logic circuits that process multiple
intracellular signals holds promise to dramatically improve the
cell type specificity of mRNA therapeutics. RNA-binding proteins

(RBPs) have often been incorporated into RNA-based synthetic circuits
in previous studies[7,16,82]. However, overexpression of widely used RBPs
such as L7Ae has been reported to increase the complexity of gene
networks and cause cytotoxicity[83]. The multi-input systems demon-
strated in this study (Fig. 7 and Fig. 8d, e) do not require RBPs for their
construction, thus making them simpler and safer to use than previous
systems[16] (Supplementary Fig. 12).

An issue with the split ON switch is that the increased amount of
OFF switches introduced potentially leads to an undesired decrease in
the ON level (activity of the output protein in the presence of target
miRNA activity) due to leakage from OFF switches. The decreased ON
level observed in Fig. 2 is likely attributable to this effect. In addition,
competitive inhibition by inactive C-terminal fragments, produced
from the leak-cancellers used in this study, is likely less efficient in OFF
level suppression compared to monomeric (1:1) inhibition, as seen in
the Barnase-Barster system[6]. This challenge could be improved by
engineering the leak-canceller itself. Engineering of the intein
sequence on the leak-canceller side[84] or the extein sequence adjacent
to intein[38,85] has been reported previously to enhance the splicing
efficiency. Optimal sequence design could enable higher leak-
cancellation efficiency with only small amounts of leak-cancellers
introduced. Furthermore, although this is a general consideration for
miRNA-responsive RNA switch technologies and not specific to our
system, it is necessary to identify at least one miRNA with cell type-
specific high (or low) activity in each target cell, and in some cases, to
optimise RNA delivery methods. For instance, the cell type-specific
disruption of the DMD exon 45 splicing acceptor shown in cultured
cells (Fig. 5d, e) holds promise for mRNA-based in vivo DMD
therapies[62]. However, for clinical translation, further optimisation will
be required regarding delivery efficiency to in vivo muscle tissue,
compatibility with existing RNA delivery technologies, and target
miRNA selection.

In conclusion, we report that *trans*-protein splicing facilitated by
split-inteins allows for the integration of outputs from RNA-based
translational control systems, thereby enabling the generation of more
desirable cellular outcomes. To our knowledge, this study represents
the first demonstration of combining mRNA-based translational reg-
ulation with post-translational protein splicing. Furthermore, we pre-
sent the first RNA-based synthetic circuit capable of detecting and
integrating signals simultaneously from distinct molecular classes,
including miRNAs and proteins. In principle, the split RNA switch
system can be adapted to any gene and implemented simply by ORF
sequence design, making it directly applicable to research and medical
applications using existing mRNA-based technologies (e.g., UTR
design, transfection, and delivery methods), as well as circular mRNA
and self-replicating RNA. Even if future optimisation of individual RNA
switches improves their performance, further enhancement in speci-
ficity may be achieved by adapting the split RNA switch approach
presented here. Split RNA switch not only represents a promising

application of protein splicing but also demonstrates the potential of post-translational processing as a comprehensive solution for existing mRNA-based gene control technologies toward practical application.

## Methods

### Preparation of modified mRNA

Splitting of the proteins used in this research was performed as shown in Supplementary Figs. 13–19. The template DNA for in vitro transcription was made by PCR with KOD One (TOYOBO, KMM-101). All mRNAs were generated using the above PCR products and MEGAscript T7 Transcription Kit (Thermo Fisher Scientific, AMB13345). To suppress immune responses and enhance translation efficiency, we used $N1$-methylpseudouridine-5′-triphosphate, mΨ (TriLink, N-1081-10), instead of uridine-triphosphate, U, and Cap Analogue, CleanCap AG or CleanCap AG (3′ OMe) (TriLink, N-7413-10, N-7413-10), except for mRNA #70, 71, which were transcribed using U. Reaction mixtures were incubated at 37 °C for up to 4 h, mixed with TURBO DNase from Transcription Kit, and further incubated at 37 °C for 30 min to remove the template DNA. The resulting mRNAs were purified using Monarch RNA Cleanup Kit (New England Biolabs, T2040L), incubated with Antarctic Phosphatase (New England Biolabs, M0289L) at 37 °C for 30 min, and then purified again using the same kit. The detailed sequence information and the set of RNA switches used in this study are shown in Supplementary Table 1 and Supplementary Table 2.

### Synthetic miRNA mimics and inhibitors

MiRNA mimics are small, chemically modified double-stranded RNAs that mimic endogenous miRNAs. mirVana miRNA Mimics (hsa-miR-21-5p, hsa-miR-302a-5p, and Negative Control #1) (Thermo Fisher Scientific, MC10206, MC12557, and 4464058) were used as mimic molecules in HEK293FT cells. The negative control mimic has a random sequence validated to have no effect on endogenous mRNAs.

MiRNA inhibitors are chemically modified, single-stranded oligonucleotides that bind specifically to and inhibit endogenous target miRNAs. mirVana miRNA Inhibitors (hsa-miR-21-5p and Negative Control #1) (Thermo Fisher Scientific, MH10206 and 4464076) were used as the miRNA inhibitors in HeLa cells.

### Culture condition

HeLa cells, HeLa cells stably expressing hmAG1 containing a nuclear localisation signal M9 (HeLa$_{hmAG1-M9}$), and HeLa cells stably expressing EGFP (HeLa$_{EGFP}$) were cultured in DMEM High Glucose medium (Nacalai Tesque, 08459-64) supplemented with 10% FBS (Foetal Bovine Serum; Biosera, FB-1003/500). HEK293FT cells (Invitrogen, R70007), HEK293FT cells stably expressing iRFP670 containing M9 (HEK293FT$_{iRFP670-M9}$), and HEK293FT cells stably expressing EGFP with DMDex45 inserted (HEK293T$_{EGxxFP}$)[57,69] were cultured in DMEM High Glucose medium supplemented with 10% FBS, 1× MEM Non-Essential Amino Acids Solution (Thermo Fisher Scientific, 11140050), 1 mM sodium pyruvate (Sigma Aldrich, S8636-100ML), and 2 mM L-glutamine (Thermo Fisher Scientific, 25030081). HeLa and HEK293FT cells were harvested and seeded as follows: after a rinse with PBS (Dulbecco's Phosphate Buffered Saline; Nacalai Tesque, 14249-24), cells were incubated with Trypsin-EDTA (0.25%) (Thermo Fisher Scientific, 25200072) for 5 min at 37 °C. After being suspended in culture medium, the necessary number of cells were dispensed for seeding.

201B7, a wild-type human iPSC line, and 201B7 stably expressing EGFP (hiPSC$_{EGFP}$, iPS_EGFP (317-12): 317-12, a heterozygous AAVS1-GFP clone in ref. 53) were cultured in StemFit AK02N (Ajinomoto, AK02N). iPSCs were harvested and seeded as follows: after a rinse with PBS, they were incubated with Accutase (Nacalai Tesque, 12679-54) for 10 min at 37 °C. The cells were then suspended in medium containing 10 μM Y-27632 (Wako, 036-24023) and centrifuged at 160 × g for 5 min at room temperature. The supernatants were discarded, and the pellets were resuspended in medium also containing 10 μM Y-27632. The

necessary number of cells was dispensed and mixed with 0.25 μg/cm² iMatrix-511 silk (nippi, 892021) for seeding. 24 h after seeding, the culture medium was replaced with medium without Y-27632.

Detailed information on the conditions of cell experiments in this study is shown in Supplementary Table 3.

### mRNA transfection

All transfections were performed using Lipofectamine MessengerMAX Reagent (Thermo Fisher Scientific, LMRNA015) according to the manufacturer's protocol. The transfection reagent was applied at scales of 50 μl and 10 μl for experiments in 24-well and 96-well plates, respectively. Opti-MEM I Reduced Serum Medium (Thermo Fisher Scientific, 31985070) was used as a buffer for MessengerMAX. The MessengerMAX reagent and Opti-MEM were mixed for 10 min. The mRNA, miRNA mimics, and miRNA inhibitors to be introduced to each well were all diluted together with Opti-MEM and then mixed with the above reagent for 5 min before adding the mixtures to the culture medium.

### Imaging and image processing

Cell images were captured using a CellVoyager CQ1 confocal quantitative image cytometer (Yokogawa Electric). Image processing was performed using an ImageJ plugin (https://github.com/yfujita-skgcat/image_converter).

### Flow cytometric analysis (24-well plate)

After a wash with 100 μl of PBS, cells were incubated with 100 μl of Trypsin-EDTA (0.25%) for 5 min at 37 °C, and 150 μl of culture medium was added. The cell suspensions were analysed using CytoFLEX S (Beckman Coulter) and 525/40 nm Bandpass with OD1 Filter for hmAG1 and 660/10 nm Bandpass Filter for iRFP670. Data was extracted from the Flow Cytometry Standard (FCS) files and analysed using FlowJo 10.8.2 software (BD Biosciences).

Only the results for Supplementary Fig. 2 were obtained using BD Accuri C6 (BD Biosciences) with an FL1 (533/30 nm) filter for hmAG1 and an FL4 (675/25 nm) filter for iRFP670. All other procedures were identical to those of the other experiments.

In the case of 96-well plates, after a rinse with 100 μl of PBS, cells are treated with 50 μl of Trypsin-EDTA (0.25%), followed by the addition of 100 μl of culture medium to collect the cell suspension.

### Cell proliferation assay (WST-1 assay)

Medium supplemented with 1/10 of WST-1 reagent (Roche Diagnostics, 05015944001) was prepared and used to replace the medium of the transfected cells. After incubation at 37 °C for 1 h, absorbance was measured at 440 nm (measurement wavelength) and 620 nm (reference wavelength) by a multimode microplate reader, Spark (Tecan). The data were subtracted by the value of blanks and normalised by the results of untreated mock-transfected samples.

### Cell proliferation assay (CellTiter-Glo Luminescent Cell Viability assay)

A volume of 100 μl of CellTiter-Glo Reagent (Promega, G7570) was added to the cell culture medium present each well in 96-well plates after equilibration for 10 min at room temperature. After mixing contents on an orbital shaker at 1000 rpm for 2 min to induce cell-lysis, the plate was incubated for 10 min at room temperature to stabilise luminescent signal, followed by additional shaking at 1000 rpm for 2 min. Luminescence was measured at 560 nm (measurement wavelength) by Spark (Tecan). The data were subtracted by the value of blanks and normalised by the results of untreated mock-transfected samples.

### Reporter assay

On the day before transfection, HEK293FT cells were seeded onto 24-well plates. 24 h after seeding, mRNAs and miRNA mimics were

transfected by Lipofectamine MessengerMAX reagent at a 50 µl scale. Fluorescence images of the cells were captured, and the samples were analysed by flow cytometry 24 h after the transfection.

## Evaluation of split ON switch coding HSV-TK

HeLa cells were seeded in 96-well plates 24 h before transfection of mRNAs and miRNA inhibitors. At the same time as transfection, the medium was replaced with medium containing ganciclovir. 24 h after the transfection, the cells were analysed by WST-1 assay.

## Evaluation of split ON switch coding antibiotics resistance genes

HeLa cells were seeded in 96-well plates 24 h before transfection of mRNAs and miRNA inhibitors. 4 h after transfection, the culture medium was changed to containing puromycin (InvivoGen, ant-pr-1), blasticidin S (InvivoGen, ant-bl-1), or hygromycin B (Nacalai Tesque, 9287). 24 h after the transfection, the cells were analysed by WST-1 assay.

## HeLa cell selection from HeLa and HEK293FT cell coculture

HeLa$_{hmAG1-M9}$ and HEK293FT$_{iRFP670-M9}$ were seeded in 96-well plates (ratio, HeLa$_{hmAG1-M9}$:HEK293FT$_{iRFP670-M9}$ = 5:1) 24 h before mRNAs transfection. Mixtures of HeLa$_{hmAG1-M9}$ and HEK293FT$_{iRFP670-M9}$ were cultured in the medium for the HEK293FT cells. 4 h after transfection, the culture medium was changed to containing blasticidin S. On day 4, fluorescence images of the cells were captured, and the samples were analysed by flow cytometry. We defined HeLa$_{hmAG1-M9}$ and HEK293FT$_{iRFP670-M9}$ as FITC-A- and APC-A-positive populations, respectively, as shown in Supplementary Fig. 7.

## Evaluation of miRNA-dependent gene editing efficiency by CRISPR-Cas9 system

On the day before transfection, cells (HeLa$_{EGFP}$, hiPSC$_{EGFP}$, HEK293T$_{EGxxFP}$) were seeded onto 24-well plates. 24 h after seeding, mRNAs, sgRNAs, miRNA mimics and miRNA inhibitors were transfected by Lipofectamine MessengerMAX reagent at a 50 µl scale. The culture medium was replaced 4, 24, and 48 h (hiPSC$_{EGFP}$), and 24 h (HeLa$_{EGFP}$, HEK293T$_{EGxxFP}$) after transfection. Fluorescence images of the cells were captured, and the samples were analysed by flow cytometry 48 h (HEK293T$_{EGxxFP}$) or 78 h (HeLa$_{EGFP}$, hiPSC$_{EGFP}$) after the transfection.

## Statistics & reproducibility

All experiments were performed with at least three independent biological replicates ($n \geq 3$) to minimise variability and ensure reproducibility. No statistical method was used to predetermine sample size. Sample sizes were determined based on common practices in molecular and cellular biology, as well as previous studies[6,16]. All key experiments were independently reproduced at least three times with consistent results. No data were excluded from the analyses. The experiments were not randomised. Treatment conditions (e.g., miRNA mimic/inhibitor and RNA switch transfection) were manually assigned to specific wells, and the well identities were known during the experiment. To minimise potential bias, all samples were processed in parallel, and transfections were performed at the same time under identical conditions. Blinding was not performed because all data were acquired using automated instruments (flow cytometer, microplate reader, and cell imager). Quantitative analyses were conducted using standardised, automated procedures (FlowJo, Excel, and ImageJ), minimising the potential for operator bias.

## Reporting summary

Further information on research design is available in the Nature Portfolio Reporting Summary linked to this article.

## Data availability

All the data supporting this study are available within the main text and the Supplementary Information. Three-dimensional structure data of proteins used in this study are available in the Protein Data Bank under accession codes 3ADF, 3TYK, 1KI2, 7K0A and 5F9R, and the AlphaFold Protein Structure Database under accession code AF-P33967-F1-v4: [https://alphafold.ebi.ac.uk/entry/P33967]. Source data are provided in this paper.

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

## Acknowledgements

We thank Dr. Kelvin K. Hui (Kyoto University) for proofreading the manuscript. We also thank Dr. Yoshihiko Fujita (Kyoto University) for giving us HeLa$_{hmAG1-M9}$ and HEK293FT$_{iRFP670-M9}$. We thank Dr. Knut Woltjen (Kyoto University) and Dr. Ryo Niwa (Kyoto University) for providing the hiPSC$_{EGFP}$ cells and for their guidance in the experiments on cell type-specific genome editing using hiPSC$_{EGFP}$. We thank Dr. Akitsu Hotta (Kyoto University) and Mr. Uikyu Bang (Kyoto University) for providing the HEK293FT$_{EGxxFP}$ and sharing information on the piggybac vector sequence and the sgRNA design strategy. We also appreciate Dr. Jun Takahashi (Kyoto University), Dr. Megumi Ikeda (Kyoto University), Dr. Hidetoshi Sakurai (Kyoto University), Dr. Shunsuke Kawasaki (Osaka University), Dr. Moe Hirosawa (Yamaguchi University), Dr. Shodai Komatsu (The University of Chicago), and Mr. Fumiya Ito (Kyoto University) for their helpful advice on the experiments. This work was supported by JSPS fellows (Grant Number JP25KJ1485 (I.A.)), JSPS KAKENHI (Grant Numbers JP20H05626 (H.S.), JP20H05701 (H.O.), JP20K12644 (H.O.), JP25H00970 (H.S.), JP25K03464 (H.O.)), Japan Agency for Medical Research and Development (AMED) (Grant Numbers JP23bm1323001 (H.S.), JP23bm1223002 (H.S.), JP23bm1123040 (H.S.)), JST CREST (Grant Number JPMJCR23B3 (H.S.)), iPS Cell Research Fund from Centre for iPS Cell Research and Application (Kyoto University) (H.O.), ISHIZUE 2023 of Kyoto University (H.O.), and Inamori Research Grants from Inamori Foundation (H.O.).

## Author contributions

I.A., H.O. and H.S. managed the project. H.O. conceived the original idea. I.A., H.O. and H.S. designed the experimental strategies. I.A. performed all experiments, except those shown in Fig. 2b–d, Fig. 3b, Supplementary Fig. 2, Supplementary Fig. 5a, and Supplementary Fig. 6c. I.A., H.O. and M.M. conducted the experiments shown in Fig. 2b–d, Fig. 3b, Supplementary Fig. 2, and Supplementary Fig. 5a. I.A. and K.H. performed the experiments in Supplementary Fig. 6c. I.A., H.O., and H.S. wrote the manuscript.

## Competing interests

H.S. owns shares in aceRNA Technologies Ltd. and is an outside director of aceRNA Technologies Ltd. The remaining authors declare no competing interests.
