## [Transparent Peer Review file · Nature Communications]

Split RNA switch orchestrates pre- and post-translational control to enable cell type-specific gene expression

Corresponding Author: Dr Hirohide Saito

Version 0:

Reviewer comments:

Reviewer #1

(Remarks to the Author)

The authors in the article "Split RNA switch: Programmable and precise control of gene expression by ensemble of pre- and posttranslational regulation" extensively study new ways of mRNA translation regulation. The research is innovative, and the manuscript is well written and easy to follow. However, there are some questions that answering would make the story more comprehensive.

1. In the OFF switch, does the presence of target miRNA lead to degradation of the whole mRNA or just inhibition of translation? Is the complementarity between the target site and the miRNA perfect or partial?
2. Does the extra sequence in the ON switch provide a binding site for decapping enzymes, leading to decapping and degradation of mRNA? Shouldn't these sequences be located on the 5' end if the above mechanism is in play? Can you provide some mRNA stability studies (for both switches)? Can you provide some comparison using a control sequence downstream of the polyA site that will not bind to Dcp2?
3. Why is the whole mRNA of the ON switch not degraded after miRNA binding to its target site that leads to cleavage?
4. What was the concentration of transfected RNA?
5. How were individual split combinations transfected? Were they complexed with the transfection agent individually, or mixed together and then complexed with the delivery carrier?
6. Do the authors know if split proteins formed with mutated parts could be toxic, and what is their level?

Reviewer #2

(Remarks to the Author)

In this manuscript, Abe et al. described a split-intein-based strategy to increase the ON/OFF ratios of the miRNA ON switch and carry out logic operations with two miRNA inputs. The authors split the target protein, which could be rejoined by protein splicing, and expressed an inactive C-terminal protein fragment as a "leak-canceller". The authors demonstrated that this strategy could improve the performance of the ON switch and achieve cell type-specific cell fate by splitting three antibiotic-resistant genes.

This work is interesting, but I would not recommend this paper for publication in Nature Communications. The reasons are listed below:

1. The conceptual innovation does not sufficiently impress me, as the designs of miRNA-responsive ON-/OFF-switches, the concept of using miR-21-5p-responsive RNA switch to distinguish HeLa/293T cells, and inducing cell type-specific cell death have already been described in their previous work (DOI: 10.1126/sciadv.abj1793). Further, the concept of basal activity reduction using split intein has been described previously (DOI: 10.1038/s41467-021-22404-9).
2. Although the authors claim that the split RNA switch could address the cell type specificity challenges hindering practical applications of RNA switches, their results (fluorescence or survival rates) remain proof-of-concept, and no real biomedical applications have been demonstrated.
3. While the logic capability of intein-based systems has been widely recognized (e.g., DOI: 10.1126/sciadv.abe9375), the authors only demonstrated three simple two-input logic gates (AND, NIMPLY, NOR) in Figure 6. Other two-input logic gates (OR, IMPLY, XOR, XNOR) and three-input logic gates could be built to evaluate the capability of the split RNA switch to assimilate multiple signals.

Minor revisions:

1. Abstract, line 11. "multi-output and multi-input RNA-based synthetic circuits" should be "two-output and two-input".
2. The format of the references needs significant modification to ensure consistency, such as standardizing journal names and paper titles.

Version 1:

Reviewer comments:

Reviewer #1

(Remarks to the Author)

All my comments were addressed.

Reviewer #2

(Remarks to the Author)

In this revised manuscript, the authors have basically addressed major criticisms through expanded experimental validation and textual clarifications. However, several issues must be addressed before considering publication in Nature Communications. The following detailed comments are given to strengthen their manuscript:

- 1 . In my assessment, the results presented in Figure 5 (DMD gene therapy) remain at the proof-of-concept stage and have not yet demonstrated true biomedical applicability. To validate the practical utility of this method, could the authors supplement the manuscript with experimental results in DMD disease mouse models or bone marrow-derived stem cell models? Such validation is critical if the authors intend to translate this approach into real-world biomedical applications.
- 2 . I note that the "Split RNA switch" described in this manuscript appears to rely entirely on pre-translational regulation (via RNA design), as the intein-mediated trans-splicing process occurs spontaneously without additional post-translational control mechanisms. This raises concerns about the accuracy of the title: "Split RNA switch: Programmable and precise control of gene expression by ensemble of pre- and post-translational regulation." The term "post-translational regulation" may be misleading unless explicit evidence of post-translational control is provided.
- 3 . The term "hmAG" in Figure 8C is inconsistent with "hmAG1" used throughout the main text. Standardization of nomenclature is essential to avoid confusion.
- 4 . The reference section contains formatting inconsistencies. For example, References 1 and 2 differ in the number of authors listed (e.g., full list vs. "et al."). Journal article titles exhibit inconsistent capitalization (e.g., some titles use sentence case while others use title case). A thorough revision to align with the journal's style guide is required.
- 5 . The current manuscript suffers from poor visual organization, including:
 - a) The graphics are too sparse and look uncoordinated.
 - b) Inconsistent font sizes and axis labeling across panels.
 - c) Redundant or overlapping annotations in schematics.
 - d) A comprehensive redesign to simplify and unify the graphical presentation is strongly recommended to enhance readability.

We sincerely thank all the reviewers for their valuable feedback and constructive comments, which have helped us greatly improve our manuscript. We have carefully addressed each point with detailed responses and made the necessary revisions. We believe that these changes have significantly enhanced the novelty, readability, and clarity of our work, and we have thoroughly addressed all the reviewers' suggestions adequately.

Point-By-Point Response:

Reviewer #1

The authors in the article “Split RNA switch: Programmable and precise control of gene expression by ensemble of pre- and posttranslational regulation” extensively study new ways of mRNA translation regulation. The research is innovative, and the manuscript is well written and easy to follow. However, there are some questions that answering would make the story more comprehensive.

Response:

We thank the reviewer for the positive feedback and valuable comments about how to improve our manuscript. We agree that the reviewer's points are very important. Thus, we revised our manuscript with additional experiments and explanations to improve its quality.

1. In the OFF switch, does the presence of target miRNA lead to degradation of the whole mRNA or just inhibition of translation? Is the complementarity between the target site and the miRNA perfect or partial?

Response:

We used a perfectly complementary sequence as the miRNA target site. Cleavage likely occurred at the miRNA target site, followed by subsequent mRNA degradation, because previous studies indicated that Argonaute 2 protein induces mRNA cleavage through the perfect complementary sequence with target miRNAs (Bartel, D. P. *Cell*. 173, 20–51 (2018)). Indeed, although the data are based on circular RNA, degradation in the presence of miRNA has been demonstrated for RNAs with perfectly complementary target sequences (Kameda, S. et al. *Nucleic Acids Res.* 51, e24 (2023)).

To clarify the above point clearly, we modified the following sentences in the revised manuscript (blue italicized part):

“While it is translated in the same manner as normal mRNA in cells without the target miRNA activity (miRNA- cells), ~~transfection~~ transgene expression from the OFF switch is inhibited by miRNA binding to the target site in cells possessing the specific miRNA activity (miRNA+ cells), likely due to the cleavage at the miRNA target site, followed by subsequent mRNA degradation (Fig. 1a, left).”

2. Does the extra sequence in the ON switch provide a binding site for decapping enzymes, leading to decapping and degradation of mRNA? Shouldn't these sequences be located on the 5' end if the above mechanism is in play? Can you provide some mRNA stability studies (for both switches)? Can you provide some comparison using a control sequence downstream of the polyA site that will not bind to Dcp2?

Response:

Thank you for your valuable comments. Since the presence or absence of “DE” does not affect the OFF level, the extra sequence is unlikely to promote decapping (Supplementary Figure 2). As the reviewer pointed out, this may be because “DE” is located at the 3' end and/or due to the introduction of nucleotide modifications. In the previous work (DOI: 10.1126/sciadv.abj1793), we reported that CAG×30 repeats are sufficient as an extra sequence for a miRNA-responsive ON switch, which exhibited a comparable ON/OFF ratio. In this manuscript, we chose “DE”-containing sequence for the split-switch system, based on the relatively high ON levels shown in Supplementary Figure 2, rather than decapping-promoting activity, which we could not confirm. To clarify this point, we have described in Supplementary Figure 2 of the original manuscript as follows: *“In this study, we used the ON switch with the “-t21-DE-aK-u5g” extra sequence, which exhibited the highest translational activity in the ON state in this assay. We prioritized the ON level of the ON switch because it was initially considered that controlling the target gene with a split ON switch might reduce the ON level of protein activity.”* I apologize for any confusion that may have arisen due our insufficient explanation.

The detailed mechanism underlying the ON switch remains unclear, and thus, the design principles for achieving higher ON/OFF ratios have yet to be established. Nonetheless, despite such limitations, our split RNA switch system significantly improves the ON/OFF ratio to a practically useful level, as shown in cell type-specific purification, genome editing, and DMD gene therapy (Figs. 3-5, new data were added in revised manuscripts), by combining translation control elements at the post-translational level, even for translation control technologies such as the miRNA-responsive ON switch, whose mechanism remains poorly understood and whose designs at the molecular level struggle to overcome the limitation of a few-fold ON/OFF ratio. Although we did not confirm the difference in the stability of the ON switch in the absence or presence of miRNA, we previously showed that the “OFF” state of the miRNA-responsive OFF switch correlates with its degradation (Kameda, S. et al. *Nucleic Acids Res.* 51, e24 (2023)). The primary objective of this study is to demonstrate the possibility of a strategy combining pre- and post-translational regulation for accurate mRNA-based gene control. Therefore, a detailed mechanistic analysis of miRNA-responsive ON switch itself is beyond the scope of this manuscript. However, we also firmly believe that such analysis is crucial for further enhancing the practical utility of miRNA-responsive ON switches, and should be pursued in future studies.

To discuss these points in the main text, we have added the following sentences to the Discussion. *“Although the mechanism of the miRNA-responsive ON switch is hypothesized to involve the ability of the extra sequence added to the 3' end to suppress gene expression in the absence of miRNA, and a perfect complementary match by the endogenous target miRNA to trigger the enzymatic removal of the extra sequence by the Argonaute 2 protein, thereby relieving the suppression, the detailed mechanism remains unclear. Therefore, a rational design strategy for the extra sequence to achieve improved ON/OFF control has not yet been established. Even though translational control technologies such as the miRNA-responsive ON switch, whose mechanism and molecular design strategy remain poorly understood and thus continue to be hindered by low ON/OFF ratios, we have illustrated significant improvements by incorporating well-studied post-translational*

control elements. As a result, our split RNA switch system improves the ON/OFF ratio to a level for practical use, as demonstrated in cell type-specific purification, genome editing, and DMD gene therapy (Figs. 3-5)."

3. Why is the whole mRNA of the ON switch not degraded after miRNA binding to its target site that leads to cleavage?

Response:

We thank the reviewer for raising this important points. Based on the performance of the ON switch, we expect that even after miRNA binding and cleavage of the extra sequence, the poly(A) tail remains at the 3' end, allowing PABP binding, which likely helps maintain the stability of the upstream region of the mRNA, thus facilitating translation in the presence of the target miRNA. In contrast, in the OFF switch, cleavage at the miRNA target site separates the upstream region containing the cap structure, leading to the loss of translational capability.

4. What was the concentration of transfected RNA?

Response:

We apologize for any confusion. The transfection amounts of mRNAs used in each experiment are provided in Supplementary Table 3.

5. How were individual split combinations transfected? Were they complexed with the transfection agent individually, or mixed together and then complexed with the delivery carrier?

Response:

To ensure a consistent concentration ratio of the introduced RNAs, all RNAs were mixed before being combined with the transfection agent to form lipoplexes.

To clarify this point, we have added the following description to the Methods:

"The mRNA, miRNA mimics, and miRNA inhibitors to be introduced to each well were all diluted together with Opti-MEM and then mixed with the above reagent for 5 min before adding the mixtures to the culture medium."

6. Do the authors know if split proteins formed with mutated parts could be toxic, and what is their level?

Response:

The inactive protein variant contains only a single amino acid mutation, so we expect toxicity comparable to the wild-type protein. To confirm this, we conducted a new CellTiter-Glo Luminescent Cell Viability assay in HeLa cells expressing either the active or inactive hmAG1, and observed no significant difference in cell viability (new Supplementary Figure 12, as shown below). In addition, those results suggest that unreacted fragments or cleaved inteins have negligible toxicity and do not require further consideration.

Supplementary Figure 12: Evaluation of cytotoxicity of split-intein and mutated protein fragments in HEK293FT cell.

Viability of HeLa cells transfected with each set of mRNAs in the CellTiter-Glo Luminescent Cell Viability Assay. The viability of each cell line was determined by dividing the values of each condition by those of the mock control. Error bars represent means \pm SD (n=3), and data of each biological replicate are shown as a point. Statistical analysis by two-sided Welch's t-test, n.s.: not significant ($P>0.05$).

Reviewer #2

In this manuscript, Abe et al. described a split-intein-based strategy to increase the ON/OFF ratios of the miRNA ON switch and carry out logic operations with two miRNA inputs. The authors split the target protein, which could be rejoined by protein splicing, and expressed an inactive C-terminal protein fragment as a “leak-canceller”. The authors demonstrated that this strategy could improve the performance of the ON switch and achieve cell type-specific cell fate by splitting three antibiotic-resistant genes.

This work is interesting, but I would not recommend this paper for publication in Nature Communications. The reasons are listed below:

Response:

We thank the reviewer for the important feedback and valuable comments for improving our manuscript. We revised our manuscript with additional experiments and explanations to highlight the conceptual advances over previous studies. We added a variety of important new experiments, including the regulation of Cas9 protein in human induced pluripotent stem cells (new Fig. 5a-c), miRNA-dependent *dystrophin* gene editing for Duchenne muscular dystrophy therapy (new Fig. 5d-e), construction of both proteins (L7Ae or LIN28A) and miRNA-responsive two-input system capable of simultaneously detecting different types of biomolecules (new Fig. 8a-c), and development of three-input synthetic circuits using two orthogonal split-intein (new Fig. 8d-e).

For clarity, we have listed the achievements of this study, including newly added experiments (blue), that were not accomplished in previous studies, as follows:

- Significant improvement of the ON/OFF ratio in the miRNA-responsive ON switch system

- Purification of target miRNA-positive cells using only antibiotics and mRNA
 - Highly specific genome editing in target miRNA-positive cells, with near-complete suppression of leaky Cas9 activity in non-target cells (new Fig. 5)
 - Construction of RNA-based synthetic circuits capable of simultaneously detecting both proteins and miRNAs (new Fig. 8)
 - Construction of RNA-based three-input synthetic circuits using two orthogonal split-inteins (new Fig. 8)
1. The conceptual innovation does not sufficiently impress me, as the designs of miRNA-responsive ON-/OFF-switches, the concept of using miR-21-5p-responsive RNA switch to distinguish HeLa/293T cells, and inducing cell type-specific cell death have already been described in their previous work (DOI: 10.1126/sciadv.abj1793).

Response:

Thank you for the valuable comments. It is true that cell type-specific cell death induction was achieved in our previous study (DOI: 10.1126/sciadv.abj1793) using the Barnase (Bn)-Barstar (Bs) system. However, in that system, an innate suppressor protein (Bs) was necessary to compensate for the low ON/OFF ratio, and additional antibiotics treatment was required to eliminate mRNA-untransfected cells, which otherwise survived as “false positives.” In contrast, the split RNA switch system we presented in this manuscript enables high ON/OFF ratio regulation without relying on corresponding suppressor proteins like Bs in the Bn-Bs system. Moreover, in certain target/non-target cell combinations, differences in sensitivity to antibiotics or cytotoxic proteins can make cell purification challenging or impossible. While the previous work can only regulate an RNase (Bn) as output for cell fate control, the split RNA switch system can control various proteins such as drug-resistance proteins (PAC, HPH, and BSR) and suicide genes (HSV-TK). This versatility of the split RNA switch system allows for simple selection using the most suitable single gene in each case, which was not feasible in the previous system. In terms of versatility, the split RNA switch system is not only capable of regulating fluorescent and cell-fate-control genes but also genome-editing enzymes (data added in new Fig. 5).

Further, the concept of basal activity reduction using split intein has been described previously (DOI: 10.1038/s41467-021-22404-9).

Response:

The strategy of competitively reducing leaky output in the OFF state using a “leak-canceller”, an OFF switch coding a mutated fragment that forms an inactive full-length mutant, is conceptually distinct from the approach described in the referenced study. The previous study focused on splitting commonly used transcription factors, such as TetR, and placing them under the control of two exogenously added small molecules (arabinose and DAPG), demonstrating that basal transcription activation by TetR could be suppressed. In contrast, our split RNA switch does not rely on increasing the number of trigger molecules to suppress the leaked output. Instead, it dramatically improves the ON/OFF ratio solely by customizing the combination of introduced mRNA sets. This feature is particularly advantageous for cell type-specific gene expression control based on endogenous marker molecules, where the repertoire of available input molecules is

inherently limited. In addition, our “split RNA switch” strategy is the first study to dramatically reduce leaky expression by combining RNA switch-based translational regulation with protein-splicing-based, post-translational regulation.

2. Although the authors claim that the split RNA switch could address the cell type specificity challenges hindering practical applications of RNA switches, their results (fluorescence or survival rates) remain proof-of-concept, and no real biomedical applications have been demonstrated.

Response:

Thank you for raising the important points to improve the quality of our manuscript. As suggested, to further demonstrate the practical utility of this technology in real biomedical applications beyond fluorescence or survival rates, we have now performed cell type-specific genome editing in HeLa cells and human induced pluripotent stem cells (hiPSCs) as an additional output (new Fig. 5b-c). Moreover, we applied our system in a therapeutic context—Duchenne muscular dystrophy (DMD) (new Fig. 5d-e). Specifically, we successfully regulated disruption of the dystrophin exon 45 splicing acceptor based on differences in miRNA activity, and significantly reduced undesired Cas9 activity in miRNA-negative cells. This study is the first demonstration of the specific induction of genome editing in target miRNA-positive cells. These results further support the real-world applicability of the split RNA switch system for accurate cell type-specific gene regulation.

3. While the logic capability of intein-based systems has been widely recognized (e.g., DOI: 10.1126/sciadv.abe9375), the authors only demonstrated three simple two-input logic gates (AND, NIMPLY, NOR) in Figure 6. Other two-input logic gates (OR, IMPLY, XOR, XNOR) and three-input logic gates could be built to evaluate the capability of the split RNA switch to assimilate multiple signals.

Response:

Thank you for your valuable comments about the logic capability of our system. As suggested, we have expanded both the “types” and the “number” of input molecules to emphasize the versatility and extensibility of split-intein-based RNA logic computation. Specifically, we constructed hybrid two-input systems sensing a protein and a miRNA as inputs, as well as three-input logic circuits using two orthogonal split inteins (Fig. 8). In fact, to our knowledge, this is the first demonstration of RNA-based synthetic circuits capable of simultaneously detecting both proteins and miRNAs by single mRNA (Fig. 8a-c).

The study referenced (DOI: 10.1126/sciadv.abe9375) constructs logic circuits such as AND, NAND, NIMPLY, IMPLY, and XOR in mammalian cells using plasmid transfection. These circuits rely on the transcriptional activity of commonly used promoters (e.g., CMV and EF1 α) as inputs and employ split-intein-mediated transcription factor assembly for logic processing. In contrast, the multi-input system reported in our study is fundamentally different, as it utilizes miRNAs as inputs and can be fully implemented in an RNA-only (i.e., synthetic mRNA delivery) form. This feature allows direct application in mRNA-based therapeutics for cell type-specific regulation based on endogenous information, rather than being limited to DNA-based circuit design. Indeed, our previous research (Matsuura, S. et al. *Nat. Commun.* 9, 4847 (2018)) demonstrated

RNA-based logic computation in mammalian cells, implementing five two-input logic gates (AND, OR, NAND, NOR, XOR) and a three-input AND gate using RNA-binding proteins (RBPs) such as L7Ae and MS2CP. However, RBPs may exhibit non-specific binding to endogenous RNAs, which could cause unexpected cytotoxicity, making them less desirable for clinical applications involving transplanted cells or in vivo delivery. To our knowledge, split inteins have not been reported to induce such cytotoxicity, also supported by our new CellTiter-Glo assay results, confirming no cytotoxic effects of our split RNA switch system (new Supplementary Figure 12). This critical point makes our approach more biologically non-invasive than RBP-based synthetic circuits.

Furthermore, among the four two-input logic gates we presented (AND, IMPLY, NOR), IMPLY is demonstrated for the first time, and our NOR gate design is significantly simplified compared to RBP-based methods (Matsuura, S. et al. *Nat. Commun.* 9, 4847 (2018)), while showing a dramatic improvement in fold change from 3-fold to 14-fold. Additionally, the AND and IMPLY gates incorporating leak cancellers allowed further tunability of the fold change, making them more customizable than the previous RBP-based systems, where OFF levels were constrained by the translation inhibition efficiency of RBPs. For these reasons, we believe strongly that our split-intein-based RNA logic computation approach surpasses existing methods in terms of safety, simplicity, and customizability. Therefore, even though we did not demonstrate all the requested two-input logic gates (OR, IMPLY, XOR, XNOR), the four two-input systems we presented (AND, IMPLY, NOR) provide strong evidence for the power of split inteins in enhancing the specificity of mRNA-based therapeutics.

Minor revisions:

1. Abstract, line 11. “multi-output and multi-input RNA-based synthetic circuits” should be “two-output and two-input”.

Response:

Thank you for your suggestion. As mentioned in response 3), we have implemented a three-input system and included the results in the new Fig. 8. Thus, we have revised the phrase to “*multi-input, two-output*” accordingly.

2. The format of the references needs significant modification to ensure consistency, such as standardizing journal names and paper titles.

Response:

Thank you for your feedback. We have revised the references to ensure consistency.

We sincerely thank all the reviewers for their valuable feedback and constructive comments, which have helped us greatly improve our manuscript. We have carefully addressed each point with detailed responses and made the necessary revisions. We believe that these changes have enhanced the novelty, readability, and clarity of our work, and we have thoroughly addressed all the reviewers' suggestions adequately.

Point-By-Point Response:

Reviewer #1

All my comments were addressed.

Response:

Thank the reviewer for the positive feedback.

Reviewer #2

In this revised manuscript, the authors have basically addressed major criticisms through expanded experimental validation and textual clarifications. However, several issues must be addressed before considering publication in Nature Communications. The following detailed comments are given to strengthen their manuscript:

1. In my assessment, the results presented in Figure 5 (DMD gene therapy) remain at the proof-of-concept stage and have not yet demonstrated true biomedical applicability. To validate the practical utility of this method, could the authors supplement the manuscript with experimental results in DMD disease mouse models or bone marrow-derived stem cell models? Such validation is critical if the authors intend to translate this approach into real-world biomedical applications.

Response:

We sincerely appreciate the reviewer's comments. First of all, we would like to emphasize that our previously reported achievement of highly accurate miRNA-positive cell type-specific fate control using the split RNA switch system (Fig. 4) represents an important strategy for cell purification before transplantation in regenerative medicine. This system also enables versatility in the choice of output genes, which has been impossible with existing technologies (Fig. 3). We believe that the precision and flexibility sufficiently demonstrate the biomedical applicability of our approach.

As we described in our previous response, we have validated the versatility of the system not only through fluorescence control (Fig. 2) and cell fate control (Fig. 3, 4), but also by extending the application to genome editing (Fig. 5b, c). As an example of a practical application, we successfully achieved cell type-specific genome editing for DMD therapy (Fig. 5d), with minimized CRISPR-Cas9 endonuclease activity in non-target cells to reduce the risks of off-target mutagenesis. Through this revision process, we have demonstrated the biomedical applicability of our split RNA switch system not only in the context of regenerative medicine but also in gene therapy.

We acknowledge that further studies, such as those suggested by the reviewer, will be important to bring "cell type-specific DMD therapy" into actual clinical use. However, these additional experiments would go beyond the initial scope of this study and substantially expand its framework. Therefore, rather than conducting these additional experiments in the current manuscript, we have addressed these important future directions in the Discussion section, as follows:

“Furthermore, although this is a general consideration for miRNA-responsive RNA switch technologies and not specific to our system, it is necessary to identify at least one miRNA with cell type-specific high (or low) activity in each target cell, and in some cases, to optimize RNA delivery methods. For instance, the cell type-specific disruption of the DMD exon 45 splicing acceptor shown in cultured cells (Fig. 5d) holds promise for mRNA-based in vivo DMD therapies. However, for clinical translation, further optimization will be required regarding delivery efficiency to in vivo muscle tissue, compatibility with existing RNA delivery technologies, and target miRNA selection.”

2. I note that the "Split RNA switch" described in this manuscript appears to rely entirely on pre-translational regulation (via RNA design), as the intein-mediated trans-splicing process occurs spontaneously without additional post-translational control mechanisms. This raises concerns about the accuracy of the title: "Split RNA switch: Programmable and precise control of gene expression by ensemble of pre- and post-translational regulation." The term "post-translational regulation" may be misleading unless explicit evidence of post-translational control is provided.

Response:

We are grateful to the reviewer for this suggestion. We would like to clarify that the term “post-translational regulation” in our title does not refer to *trans*- protein splicing reaction itself. Rather, it denotes a key feature of our system: the integration of outputs from specific RNA switches at the post-translational level, ultimately regulating protein function. In the context of the Split ON switch system (Fig. 2), this feature corresponds to the irreversible assembly of an active protein fragment translated from the ON switch with an inactive protein fragment output by an OFF switch targeting the same miRNA. This layered regulatory logic is central to the design and function of our split RNA switch system.

3. The term "hmAG" in Figure 8C is inconsistent with "hmAG1" used throughout the main text. Standardization of nomenclature is essential to avoid confusion.

Response:

We thank the reviewer for carefully pointing this out. We revised the label in Fig. 8c from "hmAG" to "hmAG1" to ensure consistency with the main text.

4. The reference section contains formatting inconsistencies. For example, References 1 and 2 differ in the number of authors listed (e.g., full list vs. "et al."). Journal article titles exhibit inconsistent capitalization (e.g., some titles use sentence case while others use title case). A thorough revision to align with the journal's style guide is required.

Response:

We appreciate the reviewer's meticulous review of the reference section. According to your comments, we re-confirmed the reference section and corrected typographic errors in references 17 and 34. However, we would like to clarify the following points.

First, regarding the number of authors listed in the references, both Reference 1 and Reference 2 include the full list of authors. We assume the reviewer may have intended to refer to References 1 and 3, which differ in the use of full author lists versus "et al." We have followed the Nature Communications referencing style, which states:

"All authors should be included in reference lists unless there are six or more, in which case only the first author should be given, followed by 'et al.'"

Accordingly, we have ensured that references comply with this guideline.

Second, as for the capitalization of article titles, we have retained the original formatting as presented by the publishers of each cited article. For instance, Reference 1 uses sentence case, and Reference 2 uses title case — this formatting reflects how the titles appear on the respective journal websites and PDF files. Therefore, the differences in capitalization are not due to oversight but are intentional, reflecting the original presentation by the source journals. In fact, a review of recent publications in *Nature Communications* shows that many articles follow this same approach to referencing, supporting the validity of our method. If any specific formatting adjustments are required, we understand that detailed instructions will be provided at the accept-in-principle stage, and we plan to make the necessary revisions at that time.

5. The current manuscript suffers from poor visual organization, including:
 - a. The graphics are too sparse and look uncoordinated.
 - b. Inconsistent font sizes and axis labeling across panels.
 - c. Redundant or overlapping annotations in schematics.
 - d. A comprehensive redesign to simplify and unify the graphical presentation is strongly recommended to enhance readability.

Response:

We are thankful for the reviewer's valuable feedback regarding the visual organization of the manuscript.

According to the *Brief guide for submission to Nature Communications*,

- *"If revisions are requested, the editor will provide detailed formatting instructions at that time."*
- *"Should your manuscript be accepted, you will receive more extensive instructions for final submission of display items."*

As such, we have prioritized scientific clarity over stylistic uniformity at this stage, and we intend to make appropriate refinements and redesigns after acceptance.

Regarding (c), we acknowledge that some annotations may appear more prominent; however, these were intentionally emphasized to highlight key aspects of the data or schematic flow. We believe these highlights aid in the reader's understanding, but we remain open to editorial guidance if any specific elements are deemed excessive or unclear.